# Fine-grained functional parcellation maps of the infant cerebral cortex

**Fan Wang[1,2†], Han Zhang[2], Zhengwang Wu[2], Dan Hu[2], Zhen Zhou[2], Jessica B Girault[3], Li Wang[2], Weili Lin[2], Gang Li[2]***

[1]Key Laboratory of Biomedical Information Engineering of Ministry of Education, Department of Biomedical Engineering, School of Life Science and Technology, Xi'an Jiaotong University, Xi'an, China; [2]Department of Radiology and Biomedical Research Imaging Center, the University of North Carolina at Chapel Hill, Chapel Hill, United States; [3]Department of Psychiatry, the University of North Carolina at Chapel Hill, Chapel Hill, United States

**Abstract** Resting-state functional MRI (rs-fMRI) is widely used to examine the dynamic brain functional development of infants, but these studies typically require precise cortical parcellation maps, which cannot be directly borrowed from adult-based functional parcellation maps due to the substantial differences in functional brain organization between infants and adults. Creating infant-specific cortical parcellation maps is thus highly desired but remains challenging due to difficulties in acquiring and processing infant brain MRIs. In this study, we leveraged 1064 high-resolution longitudinal rs-fMRIs from 197 typically developing infants and toddlers from birth to 24 months who participated in the Baby Connectome Project to develop the first set of infant-specific, fine-grained, surface-based cortical functional parcellation maps. To establish meaningful cortical functional correspondence across individuals, we performed cortical co-registration using both the cortical folding geometric features and the local gradient of functional connectivity (FC). Then we generated both age-related and age-independent cortical parcellation maps with over 800 fine-grained parcels during infancy based on aligned and averaged local gradient maps of FC across individuals. These parcellation maps reveal complex functional developmental patterns, such as changes in local gradient, network size, and local efficiency, especially during the first 9 postnatal months. Our generated fine-grained infant cortical functional parcellation maps are publicly available at https://www. nitrc.org/projects/infantsurfatlas/ for advancing the pediatric neuroimaging field.

***For correspondence:**
gang_li@med.unc.edu

†The majority of this work was done when Fan Wang was with the University of North Carolina at Chapel Hill

**Competing interest:** The authors declare that no competing interests exist.

## Editor's evaluation

The study presents a useful fine-grained functional parcellation map of the infant cerebral cortex. The data and methodology presented are solid. It will facilitate future research in neuroanatomy and neurodevelopment disorders.

## Introduction

As a prerequisite for understanding how the brain works and develops, cortical parcellation maps provide a reference for cortical area localization, network node definition, inter-subject and inter-study comparison, as well as reducing data complexity while improving statistical sensitivity and power (*Glasser et al., 2016*). To understand cortical network topology, prior work has implemented clustering-based methods to group cortical vertices into parcels based on resting-state functional connectivity (RSFC) MRI in adults (*Thomas Yeo et al., 2011*; *Power et al., 2011*). However, these methods are not suitable for generating fine-grained parcellations (>100 parcels) as they typically

result in fragmented parcels that are difficult to explain. To address this issue, recent adult cortical parcellation maps have used local gradient-based methods (*Gordon et al., 2016*; *Han et al., 2018*; *Schaefer et al., 2018*; *Wig et al., 2014*) such as the local gradient of functional connectivity (FC), to identify sharp changes in RSFC patterns across the cortical surface, which promotes the accuracy and meaningfulness of parcel boundaries.

All the abovementioned studies derived functional parcellation maps using adult data, which are not suitable for infant studies because of significant differences in brain functional organization between infants and adults (*Zhang et al., 2019a*; *Power et al., 2010*). Indeed, the dynamic development of brain FC during the first two postnatal years is important for establishing cognitive abilities and behaviors that can last a lifetime (*Li et al., 2019*; *Gilmore et al., 2018*; *Nelson et al., 2007*). However, creating infant-specific cortical functional parcellation maps has been challenging, due to difficulties in acquiring high-resolution infant brain multimodal MR images and processing these images that typically have low and rapidly changing tissue contrast across this developmental period (*Li et al., 2019*; *Gilmore et al., 2018*; *Wang et al., 2019b*). Moreover, using the abovementioned methods to generate fine-grained infant functional parcellation maps is problematic, because these methods compute the local gradient map of FC for a collection of cortical surfaces after conventional cortical folding-based registration and extensive spatial smoothing of functional features. This process essentially results in poor functional alignment across individuals due to large variations in cortical folding and functional areal boundaries. Consequently, fine-grained details of the functional architecture are often blurred or missed in the resulting parcellation maps.

Herein, we aim to generate the first comprehensive set of infant-specific, fine-grained functional parcellation maps of the cerebral cortex. To this end, we utilized a large high-resolution dataset with 1064 resting-state functional MRI (rs-fMRI) scans and 394 T1- and T2-weighted structural MRI scans from birth to 2 years of age, as part of the UNC/UMN Baby Connectome Project (*Howell et al., 2019*). To ensure accuracy, all MR images were processed using an extensively validated, advanced cortical surface-based pipeline for infant MRI (*Li et al., 2015b*). To achieve accurate cortical functional alignment across individuals, we developed a novel method that leverages subject-specific local FC gradient maps. Specifically, we use local FC gradient maps that capture fine-grained functional patterns in conjunction with local cortical folding geometry to co-register individuals. To facilitate pediatric neuroimaging studies with various scientific goals, we generated two sets of parcellation maps: one collection of maps for each representative age (3, 6, 9, 12, 18, and 24 months; age-related maps), and one map that is suitable for comparison across the 3- to 24-month age range (age-independent maps) (parcellation resources are available on NITRC [https://www.nitrc.org/projects/infantsurfatlas/] and BALSA [https://balsa.wustl.edu/study/88638]).

## Results

We created the fine-grained cortical surface-based functional parcellation maps of the infant cerebral cortex using 1064 high-resolution ($2 \times 2 \times 2$ mm$^3$) resting-state fMRI scans from 197 healthy infants, with subject demographics shown in *Table 1* and Figure 7. To capture detailed patterns of sharp

**Table 1.** Demographic information of each age group from the longitudinal dataset under study.

| Age group | Age range (days) | fMRI scans | AP scans | PA scans | Structural MRI scans (males/females) | Mean ± std gestational age (days) |
|-----------|------------------|------------|----------|----------|--------------------------------------|-----------------------------------|
| 3M | 10–144 (98.2 ± 35.1) | 104 | 52 | 52 | 50 (25/25) | 280.2 ± 7.1 |
| 6M | 145–223 (184.2 ± 22.8) | 154 | 77 | 77 | 48 (24/24) | 279.7 ± 9.6 |
| 9M | 224–318 (279.1 ± 24) | 132 | 66 | 66 | 52 (26/26) | 280.3 ± 6.8 |
| 12M | 319–410 (367.7 ± 21) | 132 | 66 | 66 | 48 (24/24) | 279.8 ± 7.8 |
| 18M | 411–591 (494.5 ± 53.2) | 222 | 111 | 111 | 80 (40/40) | 277.6 ± 7.3 |
| 24M | 592–874 (722.9 ± 66.3) | 242 | 121 | 121 | 82 (41/41) | 277.3 ± 6.2 |
| Total | 10–874 (414.9 ± 220.1) | 986 | 493 | 493 | 360 (180/180) | 278.0 ± 7.8 |

transition between cortical areas, after the conventional cortical folding-based inter-individual cortical registration, the local gradient map of cortical FC was computed on each scan of each individual and further used as a reliable functional feature for function-based registration for establishing functionally more meaningful cortical correspondences across individuals. This resulted in a remarkably detailed characterization of functional boundaries on the cerebral cortex, which was thus used to generate the infant-specific fine-grained cortical functional parcellation maps. Below we describe the following results from our work: (1) showing the improvement brought by our method; (2) introducing the *age-related* local gradient maps and the corresponding parcellations, and analyzing the variabilities of *age-related* local gradient maps; (3) introducing the *age-independent* local gradient maps and the corresponding parcellation, as well as qualitative and quantitative evaluation of the *age-independent* parcellation map; (4) investigating functional network developments based on the *age-independent* parcellation; (5) investigating parcel-wise functional development based on the *age-independent* parcellation. It is worth noting that our method generates both age-related parcellations that suit each age (3, 6, 9, 12, 18, and 24 months), and an age-independent parcellation that can be applied to images taken between birth and 24 months of age. The age-related parcellations are more suitable for cross-sectional analyses that focus only on single age groups, while the age-independent parcellation is more suitable for longitudinal and cross-sectional studies involving different ages. In this work, the age-independent parcellation is used for the longitudinal analyses in 'Age-independent functional local gradient and parcellation maps', 'Network organization and development', and 'Parcel-wise development'.

## Advantage of the proposed method

The local gradient maps of FC of 3-month infant scans generated by different methods are compared in *Figure 1*, which demonstrates the advantage of our proposed method. Specifically, *Figure 1a* shows the mean local gradient map directly computed using the RSFC-2nd as in *Gordon et al., 2016*; *Schaefer et al., 2018*. *Figure 1b* shows the mean local gradient map based on individual local gradient maps, in which we first computed a local gradient map on the RSFC-2nd of each individual and then averaged them across individuals. *Figure 1c* shows the mean local gradient map generated by the proposed method, where all individual local gradient maps are co-registered using the local gradient and cortical folding as features and then further averaged across individuals. It can be observed that major patterns of the local gradient map in *Figure 1a* are well preserved in *Figure 1c* (with some examples pointed out with white arrows), implying the meaningfulness of the local gradient patterns in *Figure 1c*. Most importantly, *Figure 1c* exhibits much more detailed and clearer patterns of the local gradient map, compared to *Figure 1a, b* especially in the temporo-occipital, parietal, and lateral prefrontal areas, indicating the advantage of performing the 2nd round of co-registration based on local gradient. Consequently, the local gradient maps generated by the proposed method can capture the detailed architecture of infant FC, while maintaining the major functional patterns, thus leading to more meaningful fine-grained functional parcellation maps. We also show the local gradient maps of three random subjects in *Figure 1d*, which further demonstrates that our method can well capture these important and detailed local gradient patterns of FC, which are usually missed by the compared methods in *Figure 1a and b*.

To test whether the local gradient maps are reproducible, we randomly divided subjects into two non-overlapping parts and computed the dice ratio between their local gradient maps after thresholding. A higher dice ratio indicates higher reproducibility. By repeating this experiment 1000 times, the overall dice ratio reaches 0.9295 ± 0.0021, indicating the high reproducibility of our results (compared to 0.71 in previous adult-based parcellations *Gordon et al., 2016*).

## Age-related functional local gradient and parcellation maps

The age-related local gradient maps are computed by averaging local gradient maps of subjects in the corresponding age groups and results at 3, 6, 9, 12, 18, and 24 months are shown in *Figure 2a*. As can be observed, the major gradient patterns are distributed symmetrically bilaterally on the cortex, as observed with the central sulcus, superior temporal gyrus, middle temporal gyrus, parieto-occipital fissure, and calcarine fissure. Nevertheless, certain local gradient patterns exhibit hemispheric differences. For example, the precentral gyrus in the right hemisphere has a higher local gradient than that

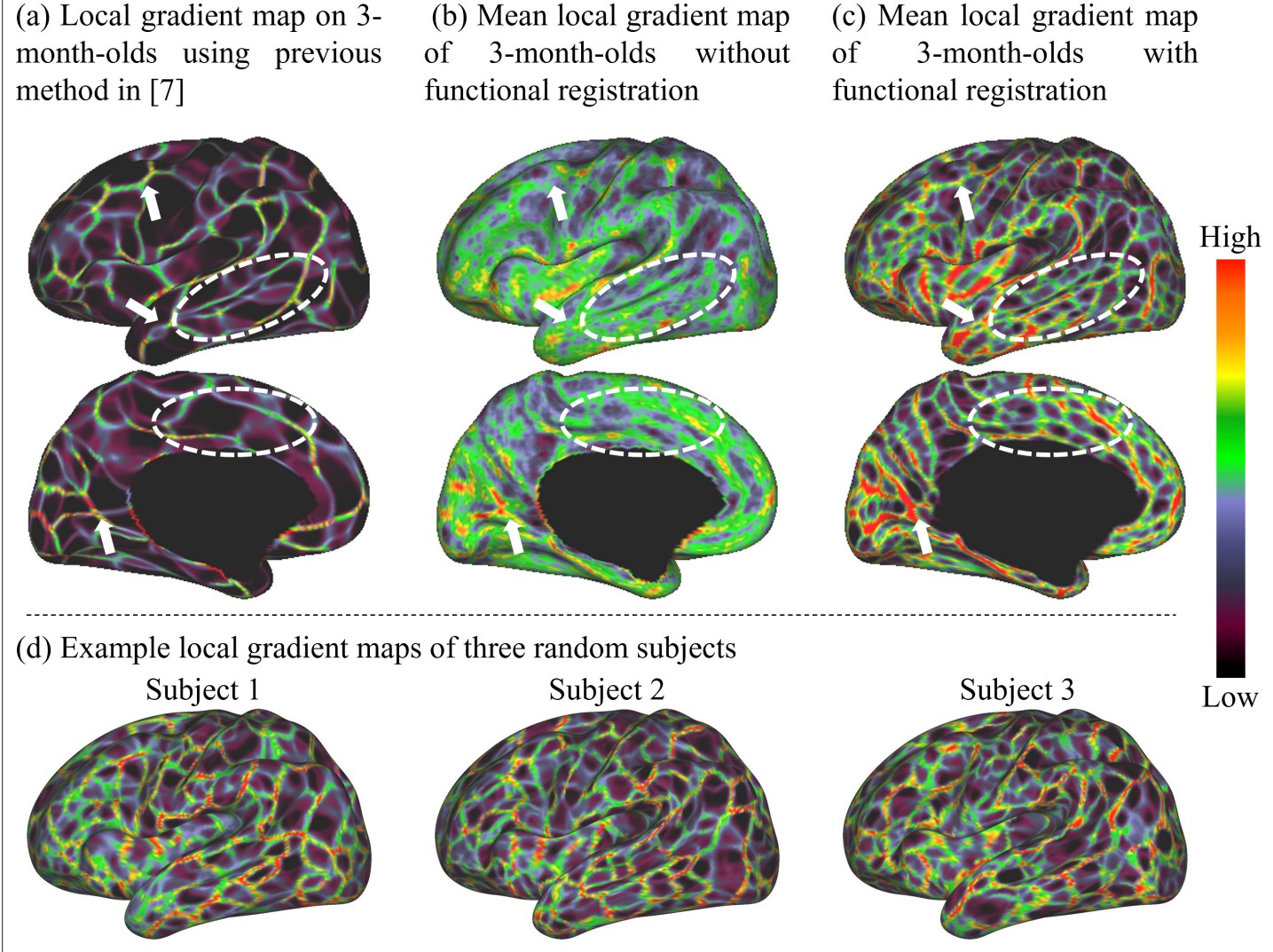

(a) Local gradient map on 3-month-olds using previous method in [7]

(b) Mean local gradient map of 3-month-olds without functional registration

(c) Mean local gradient map of 3-month-olds with functional registration

High

Low

(d) Example local gradient maps of three random subjects

Subject 1  Subject 2  Subject 3

**Figure 1.** Comparison of the mean local gradient maps of functional connectivity (FC) of the 3-month age group generated by different methods, where more detailed and clearer local gradient patterns are revealed by our proposed method. (**a**) The local gradient map computed directly on the average FC matrix across individuals as in *Gordon et al., 2016*. (**b**) The local gradient map computed on each individual and then averaged across individuals. (**c**) The local gradient map generated by our method, which computes the average of individual local gradient maps after co-registering them based on both cortical folding and local gradient features. White arrows point out consistent local gradient patterns by different methods, and white dashed circles indicate some more detailed and fine-grained patterns revealed by our method. (**d**) Local gradient maps of three random subjects. This figure demonstrates that many detailed local gradient patterns in individual cortices are usually missed by other methods, but are well captured by our method.

in the left hemisphere. All these spatial distributions of local gradient remain largely consistent across ages.

Age-related cortical parcellation maps derived from these local gradient maps are presented in *Figure 2b*. These maps were obtained using a watershed algorithm without thresholding or any manual editing. It can be observed that major local gradient patterns are well reflected as parcellation boundaries. Due to some slight differences in age-related local gradient maps, the resulting age-related parcellation maps show different parcel numbers. However, all parcel numbers fall between 461 and 493 per hemisphere, where the parcel number attains a maximum at around 9 months and then reduces slightly and remains relatively stable afterward.

To evaluate the consistency of local gradient across different age groups, we thresholded and binarized the age-related local gradient maps to their top 50% and 25% values. These binary maps were summed up, resulting in a local gradient overlap map indicating its age consistency shown in

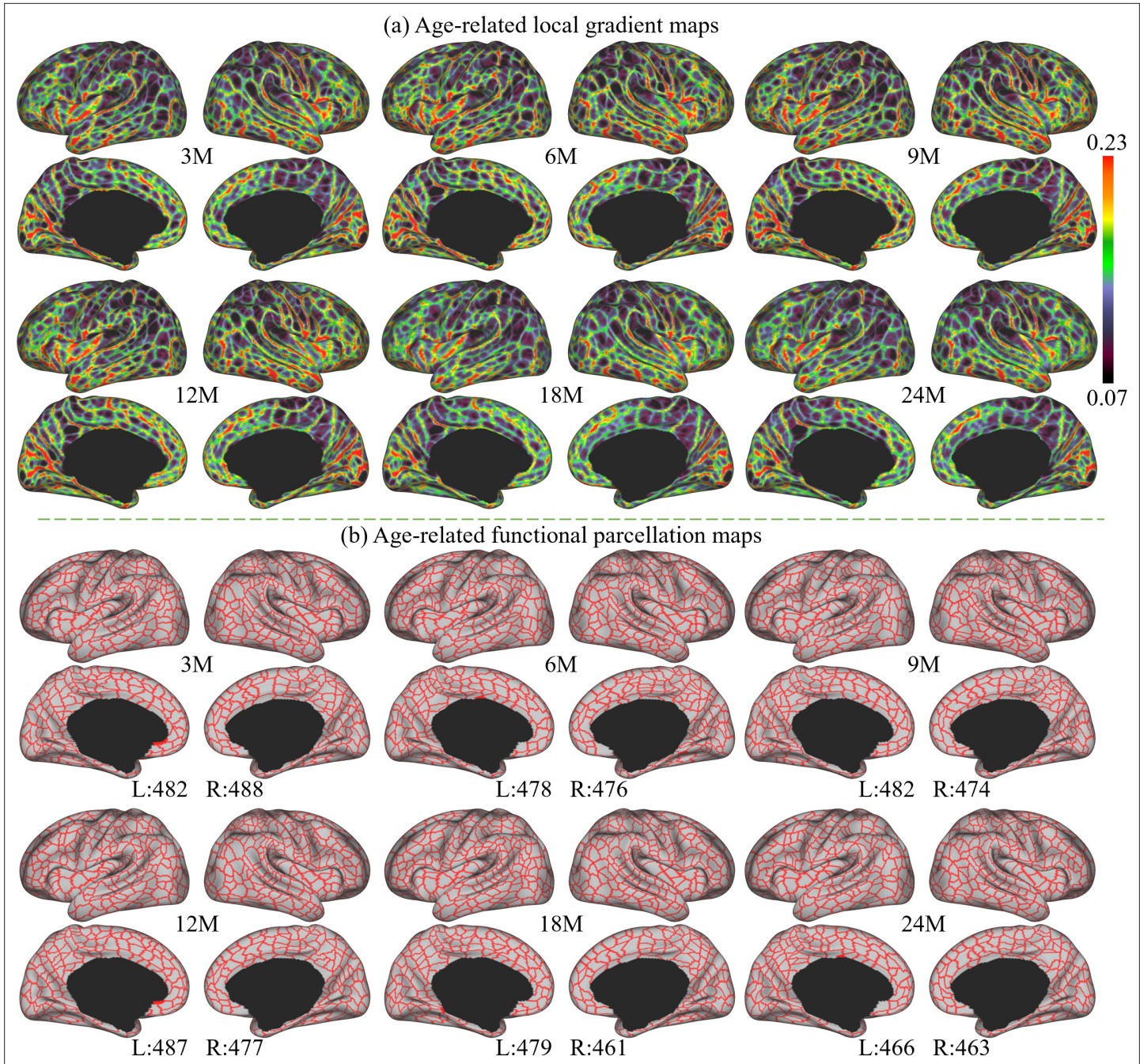

**Figure 2.** The local gradient maps and parcellation maps, where M stands for months, and 3M, 6M, 9M, 12M, 18M, and 24M represent different age groups in months defined in *Table 1*. (**a**) The age-related local gradient maps generated at different ages, where brighter colors represent higher local gradient. (**b**) Age-related fine-grained functional parcellation maps were obtained from local gradient maps using a watershed algorithm without thresholding or any manual editing. The parcel numbers are annotated below the parcellation map of each age group and are observed between 461 and 493 per hemisphere (L: left hemisphere, R: right hemisphere).

*Figure 3b*. In these maps, the colorbar represents the frequency at which the local gradient appeared across the six age groups: 'one' stands for high local gradient appeared in only one age group, and 'six' represents high local gradient appeared in all six age groups. It is worth noting that most high local gradient were repeatedly detected in all six age groups, suggesting the high consistency of majorities of high local gradient in all age groups.

Furthermore, to better illustrate the functional architecture development, we computed the across-age variability of local gradient maps between neighboring age groups, as shown in *Figure 3a*.

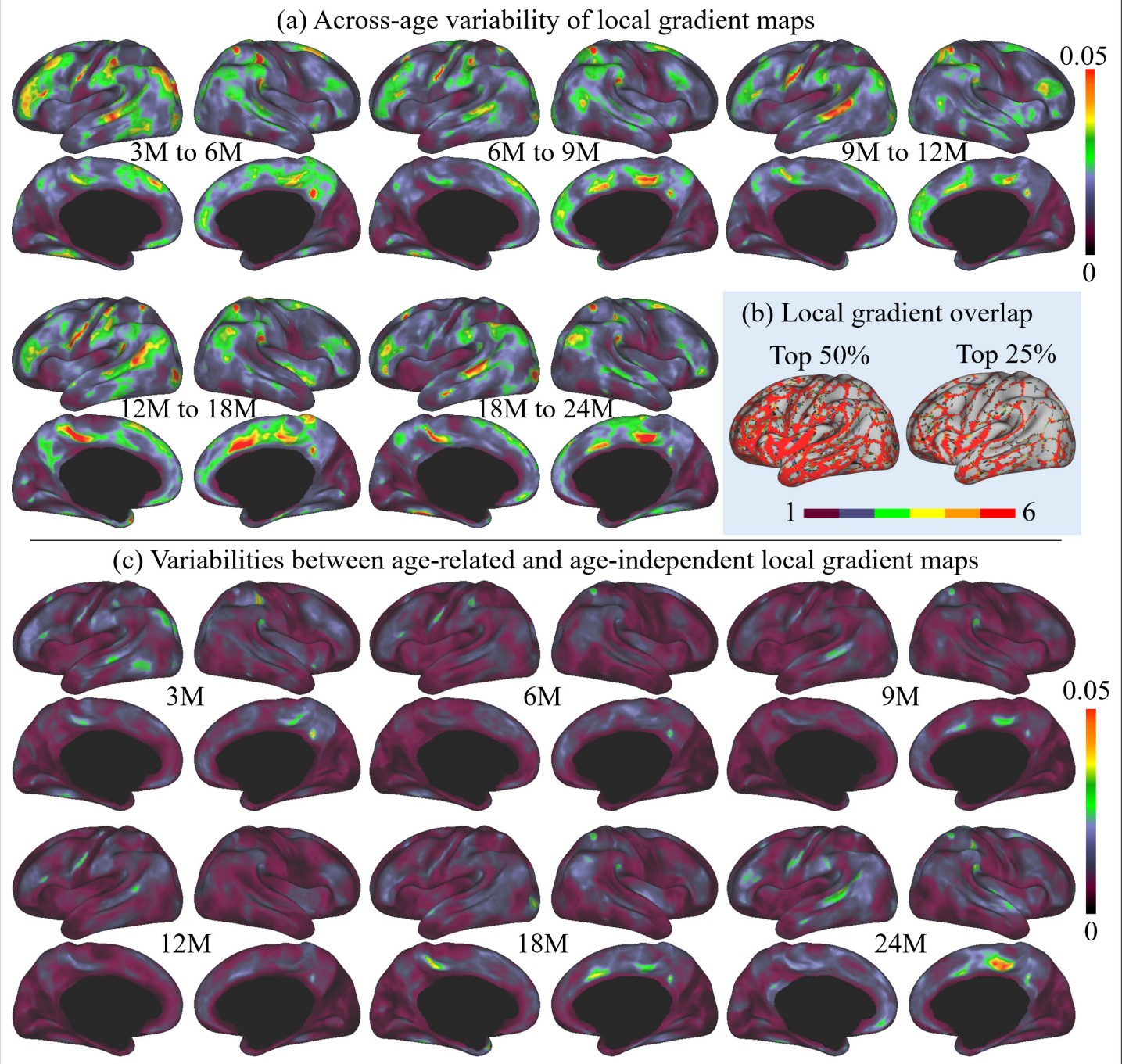

**Figure 3.** The across-age variabilities of local gradient maps and the variabilities between age-related and age-independent local gradient maps, where M stands for months, and 3M, 6M, 9M, 12M, 18M, and 24M represent different age groups in months defined in *Table 1*. (**a**) Across-age variabilities of local gradient maps between every two consecutive ages, where '3M to 6M' measures the variability of the local gradient maps between 3 and 6 months age groups and so on. The across-age variability shows a multipeak fluctuation. (**b**) Consistency of high local gradient across ages, where 'one' stands for high local gradient appeared in only one age group, and 'six' represents high local gradient appeared in all six age groups. Brighter colors stand for higher consistency of local gradient, suggesting high consistency of age-related local gradient maps. (**c**) Variabilities between each age-related local gradient map and the age-independent local gradient map.

In general, the across-age variabilities of local gradient are at a relatively low level (≤0.05) at all age intervals. Across all ages, high across-age variabilities are mainly presented in high-order association areas, including the left middle and inferior frontal, middle temporal, right superior frontal, precuneus, medial prefrontal, and bilateral supramarginal and posterior superior temporal areas. Other regions

mostly exhibit low across-age variabilities, especially in the sensorimotor and medial occipital regions. While exhibiting this overall spatial distribution, the across-age variability shows a multipeak fluctuation, where the variability decreases from 3 to 6 to 6–9 months, followed by an increase during 9–12 and 12–18 months, and drops again for 18–24 months.

## Age-independent functional local gradient and parcellation maps

Since infant functional MRI studies typically involve multiple age groups, it is highly desired to have an age-independent functional parcellation map that features parcel-to-parcel correspondences across ages, so that it can be conveniently employed for all ages during infancy. Therefore, we also computed the age-independent local gradient map (*Figure 4a*) as the average of the local gradient maps of all six age groups. The variabilities between the age-independent local gradient map and each age-related local gradient map are illustrated in *Figure 3c*. Compared to the across-age variability between neighboring age groups (*Figure 3a*), the age-independent local gradient map shows small variability to all age-related maps. The spatial distributions of high and low variabilities remain mostly similar to that of the across-age variabilities between neighboring age groups, with high variability presented in some high-order association cortices and low variability in unimodal cortices. Consequently, it can be speculated that the age-independent local gradient map can be used to generate an age-independent parcellation map that is suitable for all subjects from birth to 2 years of age.

The age-independent functional parcellation map based on the age-independent local gradient map is shown in *Figure 4b*, which has 860 parcels in total (L: 430, R: 430) excluding the medial wall. It should be noted that we manually removed some over-segmented regions after using the watershed algorithm, and before that, we had 911 parcels in total (L: 451, R: 460). The parcel boundaries of the age-independent parcellation map are well aligned with high local gradient regions and show largely bilaterally symmetric patterns of the areal organization. In the following development-related analyses in this study, we mainly employed the age-independent parcellation map to facilitate comparisons of infants across ages.

Compared to existing fine-grained parcellation maps, such as the multimodal adult parcellation (*Glasser et al., 2016*), the age-independent infant parcellation map has comparatively smaller and more evenly distributed parcel sizes and shapes. Also, as shown in *Figure 4f*, some areas of our parcels show substantial overlap with the known cortical areas of adults, such as the visual areas V1, MT, MST, sensorimotor areas 2, 3, 4, and language areas 44, 45.

To evaluate the validity of a parcellation, it is essential to compare it with a null model. This comparison not only focuses on how homogeneous or variable the parcels are, but also on whether they are more homogeneous/variable than what would be expected from parcels randomly placed with the same size and shape. Therefore, we assessed the degree to which a parcellation was more homogenous/variable than a null model consisting of many parcellations with randomly placed parcels of the same size, shape, and relative position to each other. We compared it with 1000 null parcellation maps in terms of variance and homogeneity, with the results shown in *Figure 4c, d*. It can be observed that our parcellation map shows significantly higher homogeneity (p = 2e−10) and lower variance (p = 4e−05), indicating the meaningfulness of the resulting parcellation map. We also compared our parcellation and 1000 null parcellations with the HCP parcellation using the Hausdorff distance (*Glasser et al., 2016*). From *Figure 4g*, we can observe that our parcellation generally shows statistically much lower Hausdorff distances to the HCP parcellation, suggesting that our parcellation generates parcel borders that are closer to the HCP parcellations compared to the null parcellations.

## Network organization and development

In each age group, we performed network clustering of the generated cortical parcels to reveal the early development of functional network organization. The number of functional networks for each age group is determined according to the random split-half stability analysis. Empirically, the network number is set as 2–30, and the stability plots are shown in *Figure 5a*. Empirically, it is difficult to determine the best cluster number, especially when looking for meaningful network patterns. Herein, the clustering number (network number) was chosen under several considerations. First, the stability is computed as in *Han et al., 2018*; *Fitzgibbon et al., 2020*, which is used as a reference where a higher stability is assumed to represent a better network number. Then, three empirical factors are considered: (1) for most studies, a coarse network clustering falls around 10 networks, (2) the network

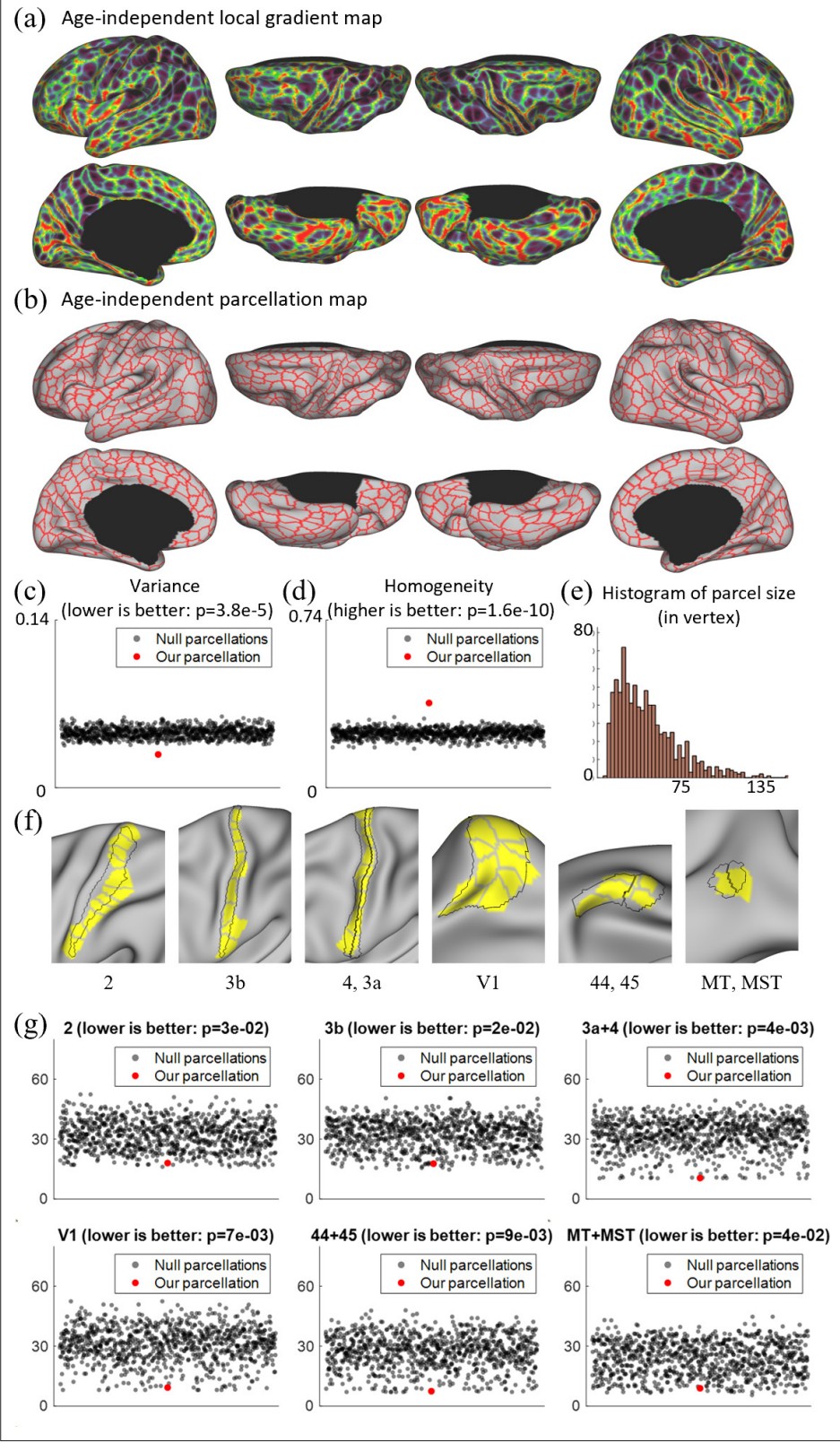

**Figure 4.** Local gradient map-derived parcels correspond to known cortical areas with low Hausdolff distances, and show lower variability and higher homogeneity than null models. (**a**) The age-independent local gradient map obtained by averaging all age-related local gradient maps. (**b**) The age-independent parcellation map obtained by applying the watershed algorithm on the age-independent local gradient map and manually removing 39 over-

*Figure 4 continued on next page*

*Figure 4 continued*

segmented regions, resulting in 860 cortical parcels, with 430 parcels in each hemisphere. (**c**) Our age-independent parcellation shows significantly lower variance compared to the null parcellations. (**d**) Our age-independent parcellation shows significantly higher homogeneity compared to null parcellations. (**e**) The histogram of parcel size, where parcel sizes are counted in vertices. (**f**) Some parcels correspond to known cortical areas defined by multimodal features in adults (*Glasser et al., 2016*). (**g**) The quantitative comparison between our parcellation and 1000 null parcellations with the Human Connectome Project (HCP) parcellation, including Brodmann's area 2, 3b, 4 + 3a, 44 + 45, V1, and MT + MST, using Hausdorff distance on the spherical space (in mm). For each circle in (**c**), (**d**), and (**g**), the *y* coordinate represents its corresponding variance, homogeneity, or Hausdorff distance, and the *x* coordinate is set to an integer between 1 and 50 in sequence for the purpose of visualization.

number is likely to grow with age, and (3) the network patterns are relatively consistent between two consecutive age groups. By considering both the stability and three empirical factors, we reviewed all clustering results between 5 and 15 clusters. As a result, we find that the most suitable cluster numbers for different age groups are 7 networks for 3 months, 9 networks for 6 months, and 10 networks for 9, 12, 18, and 24 months. Of note, we choose 10 networks for 18 months so as to be consistent during development, even though it is neither a peak nor a cliff.

The spatiotemporal patterns of the discovered functional network organization are shown in *Figure 5b*. Overall, changes in network structure from 3 to 9 months are more extensive than those from 9 to 24 months. Specifically, the sensorimotor network splits into two subnetworks from 3 to 6 months, and the boundary between them moved toward the ventral direction from 6 to 9 months. The hand sensorimotor (*Power et al., 2011*; *Manza et al., 2020*) expands, while the mouth sensorimotor (*Power et al., 2011*; *Manza et al., 2020*) shrinks, and both stabilize after 9 months. The auditory network is distinguished at 3 months and merges into the hand sensorimotor at 6 months. The visual network splits into peripheral and central visual subnetworks from 6 to 9 months and remains stable until a slight shrinkage at 24 months.

Other networks exhibit more complex development with multipeak fluctuation of the size in certain networks. Specifically, the anterior default mode network expands from 3 to 6 months, shrinks from 6 to 9 months and from 12 to 18 months, and expands thereafter. The lateral posterior default mode network that emerged at 6 months shrinks from 6 to 9 months and then expands from 9 to 18 months; while the medial posterior default mode network that emerged at 9 months only slightly shrinks thereafter. The anterior and posterior default mode networks are detected at 6 months and remain separated till 24 months, while the shape of these two networks develop to the adult-like pattern at 18 months. The superior temporal network shrinks from 3 to 6 months, expands from 6 to 9 months, and then shrinks again from 9 to 24 months. The anterior frontoparietal network undergoes an overall shrinkage from 3 to 24 months, except for a transient expansion from 12 to 18 months. On the medial surface, the posterior frontoparietal network expands to include the parahippocampal gyrus from 3 to 6 months and then disappears by 9 months. On the lateral surface, the posterior frontoparietal network expands from 3 to 9 months to include the inferior temporal part and becomes stable thereafter. The dorsal attention network is seen at 6 months and evolves to the adult-like pattern at 9 months and remains stable thereafter.

*Figure 5c* shows the reproducibility of parcels, which represents how often each parcel is allocated to the same networks in bootstrapping. The results show that most parcels are repeatedly categorized into the same network, while those moving (yellowish) parcels are distributed mostly between the boundaries of two networks. This result demonstrates the robustness and meaningfulness of the discovered network pattern.

## Parcel-wise development

Homogeneity of FC can be used as a criterion for characterizing functional development. *Figure 6a* shows the parcel-wise homogeneity development during infancy. Our results suggest that the overall parcel-wise homogeneity shows a monotonic decrease trend during the first 2 years by maintaining similar relative spatial distribution. Higher homogeneities are in the sensorimotor, paracentral, posterior insula, inferior parietal, posterior superior temporal, lateral occipital, and occipital pole. Low homogeneities are presented in lateral prefrontal, medial frontal, anterior insula, inferior temporal, and temporal pole.

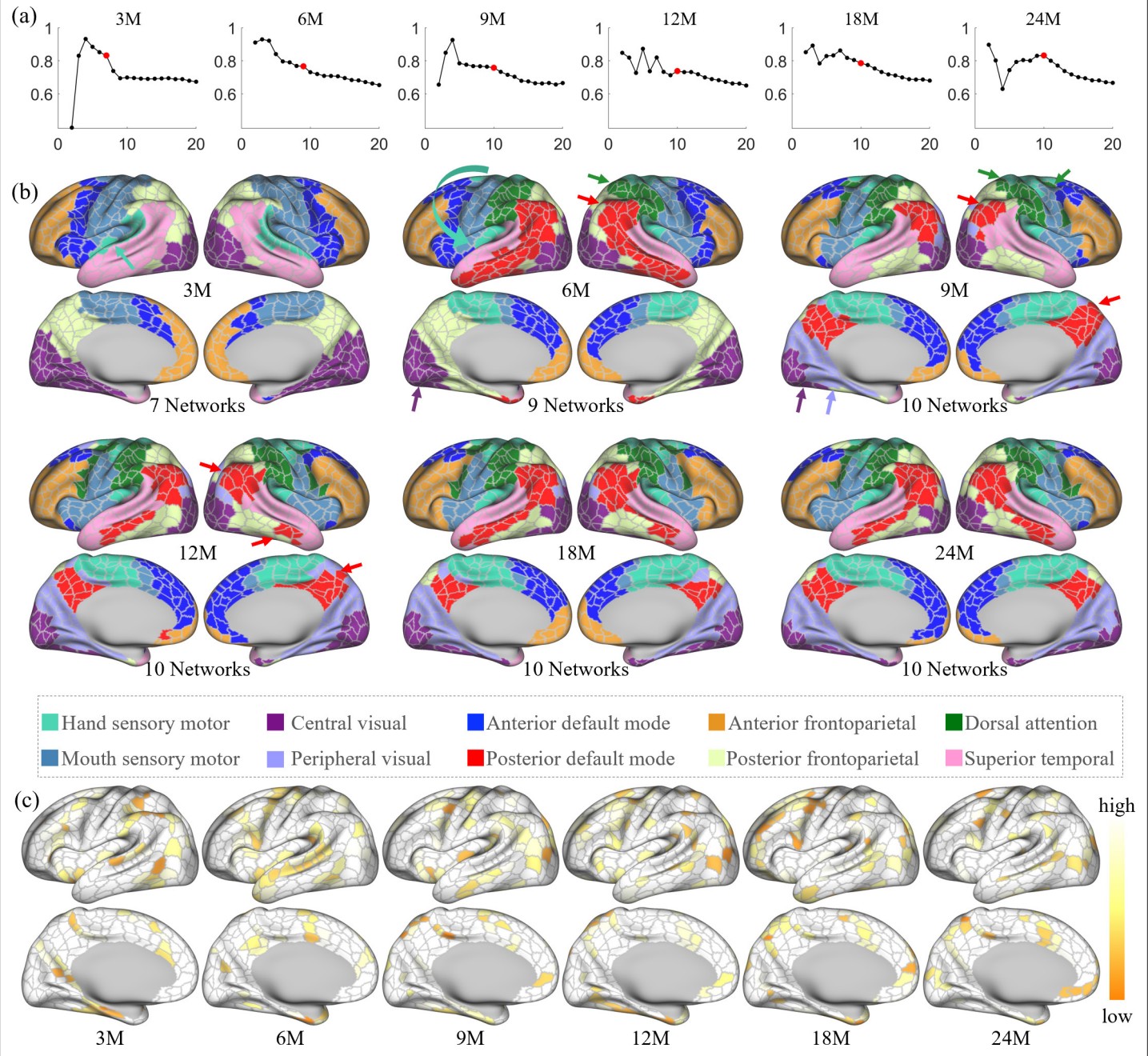

**Figure 5.** The spatiotemporal network developmental trend of infant brain functional networks. Herein, M stands for months, and 3M, 6M, 9M, 12M, 18M, and 24M represent different age groups in months defined in *Table 1*. (**a**) Stabilities of different network numbers of different age groups computed by repeating 200 times random split-half test. The selected numbers are highlighted in solid red dots. (**b**) Discovered functional network organization of parcels during infantile brain development, color coded by corresponding networks denoted below. The corresponding network numbers are annotated below each age group accordingly. Arrows point out the noteworthy changes in networks during development, colored according to the related networks. (**c**) The reproducibility of each parcel belonging to the same clustering during random clustering, where high reproducible parcels are whiter, and low reproducible parcels are more yellowish.

*Figure 6b* shows the development of the local efficiency of each parcel. Overall, the local efficiency also exhibits a strong multipeak fluctuation, with inflection ages observed at 9 and 15 months. Parcels with low efficiency are located in the lateral superior frontal, medial superior frontal, orbitofrontal, ventral insula, and anterior inferior temporal cortices. Parcels with high local efficiency are mainly observed in sensorimotor, paracentral, parietal, and precuneus regions.

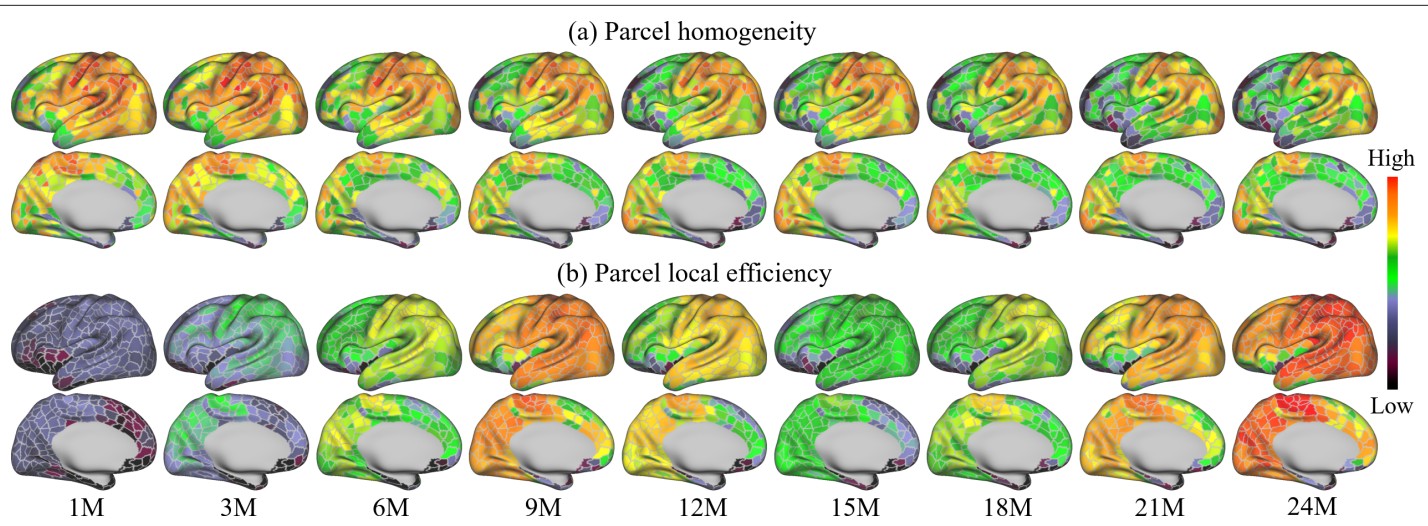

**Figure 6.** Parcel-wise development maps of infant brain functional architecture. (**a**) shows the parcel-wise homogeneity monotonically decreases with age, and (**b**) indicates that the efficiency of each parcel shows multipeak fluctuation.

## Discussion

In this study, we created the first set of both age-related and age-independent, infant-specific, fine-grained, and cortical surface-based functional parcellation maps based on local gradient maps of FC. We analyzed the spatiotemporal patterns of age-related local gradient maps and found that age-independent local gradient maps are suitable for creating fine-grained functional parcellation maps for all ages in the infant cohort. We validated the meaningfulness of the parcellation and showed that its boundaries substantially reproduced known areal boundaries, and its parcels featured high homogeneity and low variance. Finally, we illustrated the infantile development in functional network structure, parcel homogeneity, and parcel local efficiency.

This study used the local gradient of FC as a feature for improving functional alignment across individuals, in addition to the conventional cortical folding features used by previous adult functional parcellations (*Gordon et al., 2016*; *Schaefer et al., 2018*). As a result, our method not only captured important coarse gradient patterns discovered by previous methods, but also revealed much more detailed areal boundaries at a remarkable resolution, as illustrated in *Figure 1*. The main reason is that previous studies relied solely on cortical folding-based registration, and thus inevitably suffered from significant inter-subject variability in the relation between cortical folding and functional areas, leading to less accurate inter-subject functional correspondence. For infant-specific functional parcellations, the only one currently available is the volumetric-based parcellation based on image registration and clustering generated by *Shi et al., 2018*, without any advanced surface-based processing and registration. In contrast, our parcellation maps are generated based on the cortical surface, which well respects the topology of the convoluted cerebral cortex, and avoids mixing signals from opposite sulcal banks and different tissues, leading to more accurate functional signals resampling, smoothing, computation, and registration. Moreover, our parcellations leveraged high-quality 2 mm isotropic fMRI data that densely covers the first 2 years, instead of data with a coarse resolution of 4 mm isotropic centering at birth, 1 and 2 years of age (*Shi et al., 2018*).

Standing upon the detailed local gradient patterns by the proposed method, we generated age-related fine-grained parcellation maps for 3, 6, 9, 12, 18, and 24 months of age. We found that the across-age variability (temporal changes) of the local gradient generally decreases during most age intervals, except for a slight increase from 9 to 12 months. This may suggest that the development of functional architecture gradually slows down during the first 2 years. We also found that high across-age variabilities mostly presented in high-order association cortices, implying that they are developing at a more considerable pace compared to unimodal cortices. In addition, to the age-related parcellations, we also generated one age-independent parcellation on the grounds that studies typically involve subjects of different ages. Since our age-independent map shows low variability to all age-related local gradient maps, we consider it representative across the infancy period.

We will make both the age-related and age-independent functional parcellation maps accessible to the public so that researchers may select the parcellation maps that best suit their needs.

This study successfully augmented the resolution of the existing cortical parcellations from ~400 to ~900 areas, which represents a finer architecture of brain functional organizations compared to previous ones. The main reason that our method generates much finer parcellation maps is that both the functional registration and parcellation process are based on the local gradient, which characterizes a fine-grained feature map based on fMRI. This leads to both better inter-subject alignment in functional boundaries and finer region partitions. This fine-grained cortical organization is in line with *Eickhoff et al., 2018*, where they believe that 200–300 areas are not the ultimate resolution for cortical parcellations due to the hierarchical nature of the brain organization. *Glasser et al., 2016* also consider 360 as a lower bound for cortical parcellations since each parcel can be represented as a combination of several smaller regions. Even without performing surface registration based on fine-grained functional features, recent adult fMRI-based parcellations greatly increased parcel numbers, such as up to 1000 parcels in *Schaefer et al., 2018*, 518 parcels in *Peng et al., 2023*, and 1600 parcels in *Zhao et al., 2020*. Meanwhile, our strategy is different from *Glasser et al., 2016*, which jointly consider multimodal information for defining parcel boundaries, thus parcels revealed purely by functional MRI might be ignored in the HCP parcellation.

For infants, though the infant FC might not be as strong as in adults, it is believed (*Gao et al., 2015b*; *Gao et al., 2009*; *Keunen et al., 2017*) that the basic units of functional organization are likely to present in infant brains, and brain functional development gradually shapes the brain networks. Therefore, the functional parcel units in infants could be possibly on a comparable scale to adults. Consequently, our fine-grained infant cortical parcellation maps provide a great platform for analyzing the increasingly larger cohorts and higher resolution pediatric neuroimaging data, thus leading to higher accuracy for downstream analyses, such as the infant functional connectome fingerprinting (*Hu et al., 2022*). However, it is worth noting that averaging across individuals inevitably introduces some registration errors into the parcellation, especially for regions with high topographic variation across subjects (*Glasser et al., 2016*; *Glasser et al., 2013*), which would lead to loss of granularity in these regions. We believe this is an important issue that exists in most methods on population-level parcellations, and the eventual solution might be individualized parcellation.

It is worth noting that our parcellation increased the resolution in a meaningful way. First, our local gradient maps are highly reproducible. By separating subjects into non-overlapping subsets, their local gradient patterns are repeated with a dice ratio of ~0.93. Second, our age-independent infant parcellation shows high concordance with some specific cortical areas defined by *Glasser et al., 2016*, which is recognized as the state-of-the-art adult parcellation map. As illustrated in *Figure 4d*, our local gradient map-derived parcellation contains parcels that have substantial overlap with the known adult area V1 defined by *Glasser et al., 2016*. Other known cortical areas of adults, such as sensorimotor areas 2, 3, and 4, were also overlapped with a combination of several parcels in our parcellation. These observations that parcel borders conform to some adult cortical areas lend substantial visual validity to the parcellation.

When applying parcellation as a tool to explore infant brain functional development, our results reveal complex multipeak fluctuations in several aspects, including the across-age variation of local functional gradients, network organization, and local efficiency. To the best of our knowledge, this complex development trend is not reported in previous literature and fills an important knowledge gap for infant brain functional development. These functional developmental patterns are very different from early brain structure development, where cortical thickness follows an inverted-'U' shaped trajectory (*Wang et al., 2019a*), while surface area and cortical volume monotonically increase during infancy (*Huang et al., 2022*). The multipeak fluctuations potentially mirror different milestones of behavioral/cognitive abilities, which likely emerge at different ages during infancy (*Gao et al., 2015a*). However, the underlying mechanisms of such developmental patterns remain to be further investigated. Further studies and data are necessary to reproduce these patterns, as the BCP is designed as an accelerated longitudinal study, where subjects at different timepoints typically are not the same, which might induce inter-subject variabilities and affect the development patterns.

For network organization (*Figure 5b*), at 3 months, networks tend to group vertices with close spatial locations, resulting in networks being more dependent on the local anatomy. After 9 months, the primary sensory networks demonstrate adult-like patterns, while higher-order functional networks

still show substantial differences compared to the adult-like pattern. Our results suggest that a primitive form of brain functional networks is present at 3 months, which is largely consistent with recent studies suggesting that most resting-state networks are already in place at term birth (*Gao et al., 2015b*; *Gao et al., 2009*; *Keunen et al., 2017*; *Doria et al., 2010*; *Fitzgibbon et al., 2020*). Furthermore, our results also suggest that, compared to higher-order functional networks, the primary sensory system is more developed in infants. This confirms previous findings in infant cortical thickness development (*Wang et al., 2019a*), suggesting that the primary functional systems develop earlier than higher-order systems.

At the network level, the sensorimotor system splits into two subnetworks, that is, the mouth- and hand sensorimotor (*Power et al., 2011*; *Manza et al., 2020*) at 6 months, which is also observed in infants and toddlers (*Eggebrecht et al., 2017*). The visual network is split into central (primary) visual and peripheral (higher-order) visual cortices at 9 months and maintains this pattern until 24 months. This subdivision of mouth- and hand-sensorimotor networks is also found in adults (*Thomas Yeo et al., 2011*). The higher-order functional systems, including the default mode, frontoparietal, and dorsal attention network, exhibit considerable development from 3 to 9 months, followed by some minor fine-tuning from 12 to 24 months. A previous infant study (*Gao et al., 2009*) also demonstrated that functional network development shows more considerable change in the first year compared to the second year. At 24 months, both default mode and frontoparietal networks show a lack of strong cross-lobe connections. Though several studies identified some prototypes of cross-lobe connection (*Gao et al., 2009*; *Shimony et al., 2016*), their links have not been reliably distinguished (*Grayson and Fair, 2017*; *Dosenbach et al., 2010*; *Kelly et al., 2009*). Our results suggest that the higher-order functional networks are far more from established at 24 months of age. It is worth noting that the change in cortical area assigned to each network can be quite subtle within a short time interval, which emphasizes the importance of using a fine-grained parcellation map.

The parcel homogeneity measures the development within parcels. Our result shows (*Figure 6a*) that unimodal cortices, including the sensorimotor, auditory, and visual areas, show high homogeneity, which is largely consistent with adults (*Gordon et al., 2016*). However, the inferior parietal and posterior superior temporal cortices, which show high homogeneity in infants, are observed low homogeneity in adults (*Gordon et al., 2016*). Furthermore, the prefrontal area, which shows relatively low homogeneity in infants, seems to develop a moderate-to-high degree of homogeneity in adults. Almost all parcels are observed to decrease homogeneity with age. This is likely related to the development of brain function, especially in higher-order cortices, which show increased heterogeneity, and consequently decreased homogeneity. Among the higher-order association cortices, the prefrontal area has the lowest homogeneity, followed by temporal and then parietal regions, suggesting different levels of functional development.

Local efficiency measures a different aspect of development – it represents the connection of parcels to neighbors. Higher-local efficiency is usually related to higher functional segregation (*Zhang et al., 2019a*). Our results (*Figure 6b*) suggest that local efficiency shares certain similar spatial distributions with homogeneity – they both increase in anterior to posterior (AP) and ventral to dorsal directions. During development, the local efficiency shows a complex developmental trend: although 24 months show a strong increase compared to 3 month, there is a dip from 12 to 21 months that should be noted. The age-related increase in local efficiency was previously found from 18 months to 18 years (*Hagmann et al., 2010*), 5–18 years (*Wu et al., 2013*), and from 12 to 30 years (*Dennis et al., 2013*), and is likely explained by progressive white matter maturation (*Hagmann et al., 2010*). This trajectory of local efficiency is not contradictory to the previous studies (*Jiang et al., 2019*; *Gao et al., 2011*), since they only measured the averaged local efficiency of all nodes to reflect network characterization, thus missing important characteristics of parcel-level local efficiency. This further stresses the importance of performing parcel-wise analyses and the significance of fine-grained infant cortical parcellations.

## Conclusion

In summary, for the first time, this study constructed a comprehensive set of cortical surface-based infant-specific fine-grained functional parcellation maps. To this end, we developed a novel method for establishing functionally more accurate inter-subject cortical correspondences. We delineated age-related parcellation maps at 3, 6, 9, 12, 18, and 24 months of age as well as an age-independent

parcellation map to facilitate studies involving infants at different ages. Our parcellation maps were demonstrated to be meaningful by careful comparison with known areal boundaries and through quantitative evaluation of homogeneity and variance of FC. Leveraging our infant parcellation, we provide the first comprehensive visualizations of infant brain functional developmental maps on the cortex, which will serve as valuable references for future early brain developmental studies. Our generated fine-grained infant cortical functional parcellation maps are publicly available at https://www.nitrc.org/projects/infantsurfatlas/.

## Methods

### Subjects and image acquisition

Subjects in this study are from the UNC/UMN Baby Connectome Project (BCP) dataset (*Howell et al., 2019*). The BCP focuses on normal early brain development, where all infants were born at the gestational age of 37–42 weeks and free of any major pregnancy and delivery complications. In this study, 394 high-resolution longitudinal structural MRI scans were acquired from 197 (90 males and 107 females) typically developing infants, as demonstrated in *Figure 7*. Images were acquired on 3T Siemens Prisma MRI scanners using a 32-channel head coil during natural sleeping. T1-weighted images (208 sagittal slices) were obtained by using the three-dimensional magnetization-prepared rapid gradient echo (MPRAGE) sequence: TR (repetition time)/TE (echo time)/TI (inversion time) = 2400/2.24/1600 ms, FA (flip angle) = 8°, and resolution = $0.8 \times 0.8 \times 0.8$ mm$^3$. T2-weighted images (208 sagittal slices) were acquired with turbo spin-echo sequences (turbo factor = 314, echo train length = 1,166ms): TR/TE = 3200/564 ms, and resolution = $0.8 \times 0.8 \times 0.8$ mm$^3$ with a variable flip angle. All structural MRI data were assessed visually for excessive motion, insufficient coverage, and/or ghosting to ensure sufficient image quality for processing.

For the same cohort, 1064 high-resolution rs-fMRI scans were also acquired using a blood oxygenation level-dependent (BOLD) contrast sensitive gradient echo echo-planar sequence: repetition time = 800 ms, echo time = 37 ms, flip angle = 80°, field of view = $208 \times 208$ mm, 72 axial slices per volume, resolution = $2 \times 2 \times 2$ mm$^3$, total volumes = 420 (5 min 47 s). The rs-fMRI scans include 524 AP scans and 540 posterior to anterior (PA) scans, which are two opposite phase-encoding directions for better correction of geometric distortions.

### Structural MRI processing

All T1- and T2-weighted MR images were processed using an infant-specific pipeline detailed in *Li et al., 2014a*; *Li et al., 2015b*; , which has been extensively validated in many infant studies (*Wang et al., 2019a*; *Li et al., 2013*; *Li et al., 2015a*; *Geng et al., 2017*; *Meng et al., 2018*; *Li et al., 2014b*; *Jha et al., 2019*; *Lyall et al., 2015*; *Meng et al., 2016*). The processing procedure includes the following main steps: (1) Rigid alignment of each T2-weighted image onto its corresponding T1-weighted image using FLIRT in FSL (*Jenkinson et al., 2002*; *Jenkinson and Smith, 2001*; *Smith et al., 2004*; *Woolrich et al., 2009*; *Popescu et al., 2012*); (2) Skull stripping by a deep learning-based method (*Zhang et al., 2019b*), followed by manual editing to ensure the clean skull and dura removal; (3) Removal of both cerebellum and brain stem by registration with an atlas; (4) Correction of intensity inhomogeneity using the N3 method *Sled et al., 1998*; (5) Longitudinally consistent segmentation of brain images as white matter, gray matter, and cerebrospinal fluid using an infant-dedicated deep learning-based method *Wang et al., 2018*; and (6) Separation of each brain into left and right hemispheres and filling non-cortical structures.

### Resting-state fMRI processing

Infant rs-fMRI processing was conducted according to an infant-specific functional pipeline (*Jiang et al., 2019*; *Zhang et al., 2019c*; *Zhou et al., 2019*). The head motion was corrected using FSL (*Jenkinson et al., 2002*). For the spatial distortion, we performed echo planar imaging (EPI) geometric distortion corrections for AP and PA scans using *topup* from FSL (*Smith et al., 2004*) and verified the distortion correction results by visual inspection. The rs-fMRI scans were then registered to the T1-weighted structural MRI of the same subject using a boundary-based registration approach (*Greve and Fischl, 2009*; *Griffanti et al., 2014*; *Salimi-Khorshidi et al., 2014*). All of the transformations and deformation fields were combined and used to resample the rs-fMRI data in the native space through

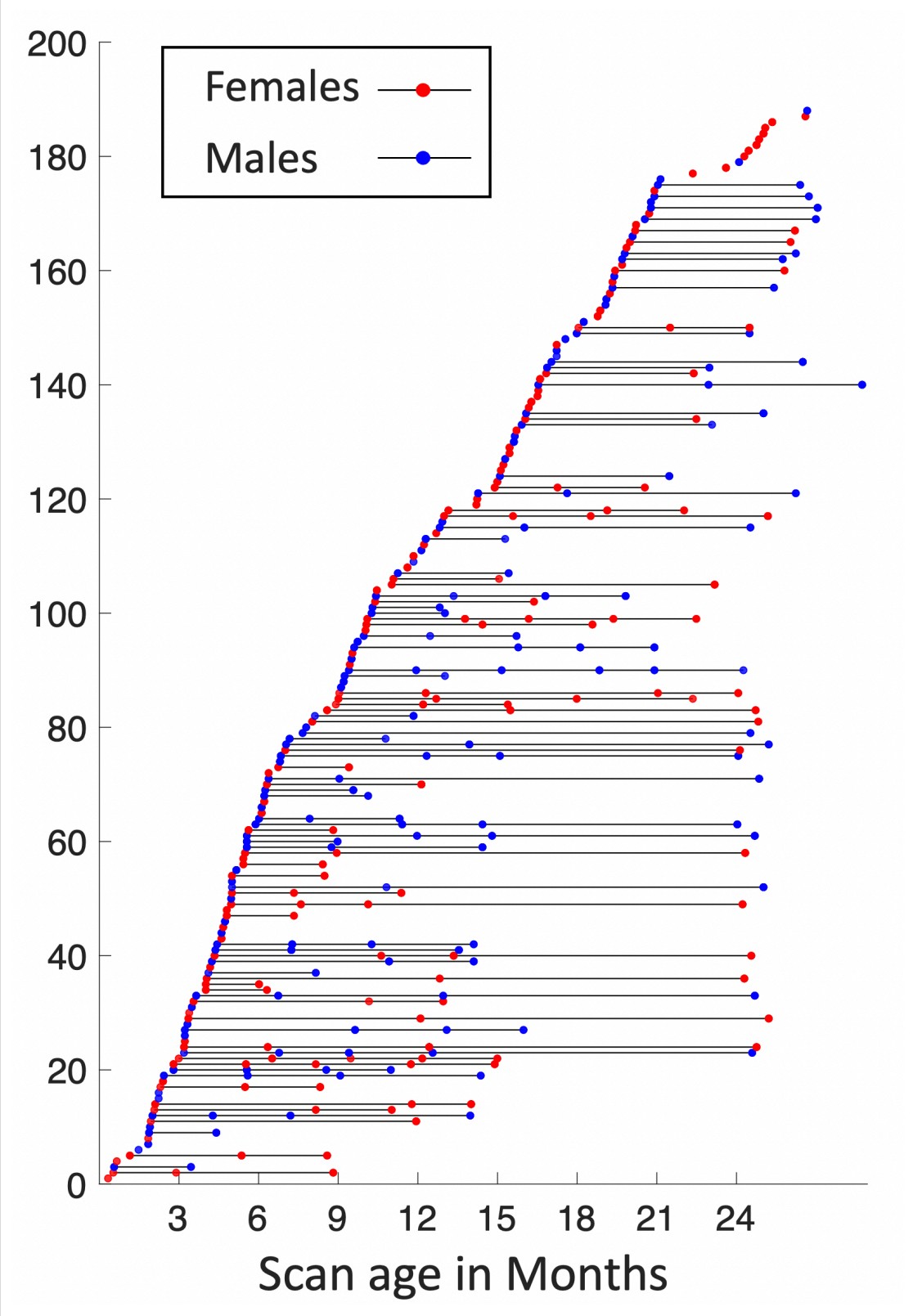

**Figure 7.** Longitudinal distribution of scans. Each point represents a scan with its scanned age (in months) shown on the x-axis, with males in blue and females in red, and each horizontal line represents one subject, with males in blue and females in red.

a one-time resampling strategy. After conservative high-pass filtering with a sigma of 1000s to remove linear trends in the data, individual independent component analysis was conducted to decompose each of the preprocessed rs-fMRI data into 150 components using MELODIC (*Beckmann and Smith, 2004*) in FSL. An automatic deep learning-based noise-related component identification algorithm was used to identify and remove non-signal components to clean the rs-fMRI data (*Kam et al., 2019*).

## Cortical surface reconstruction and mapping

Based on the tissue segmentation results, the inner, middle, and outer cortical surfaces of each hemisphere of each MRI scan were reconstructed and represented by triangular meshes with correct topology and accurate geometry, by using a topology-preserving deformable surface method (*Li et al., 2014a*; *Li et al., 2012*). Before cortical surface reconstruction, topology correction on the whiter matter surface was performed to ensure the spherical topology of each surface (*Sun et al., 2019*). After surface reconstruction, the inner cortical surface, which has vertex-to-vertex correspondences with the middle and outer cortical surfaces, was further smoothed, inflated, and mapped onto a standard sphere (*Fischl et al., 1999*).

To ensure the accuracy of longitudinal analysis during infancy, it is necessary to perform longitudinally consistent cortical surface registration (*Li et al., 2015b*). Specifically, (1) for each subject, we first co-registered the longitudinal cortical surfaces using Spherical Demons (*Yeo et al., 2010*) based on cortical folding-based features, that is, average convexity and mean curvature. (2) Longitudinal cortical attribute maps were then averaged to obtain the intra-subject mean surface maps. (3) For each hemisphere, all intra-subject mean surface maps were then co-registered and averaged to get the population-mean surface maps. (4) The population-mean surface maps were mapped to the HCP 164k 'fs_LR' space using the deformation field that deforms the 'fsaverage' space to the 'fs_LR' space released by *Van Essen et al., 2012*, which was obtained by landmark-based registration. By concatenating the three deformation fields of steps 1, 3, and 4, we directly warped all cortical surfaces from individual scan spaces to the HCP 164k_LR space and then resampled them to 32k_LR using the HCP pipeline (*Glasser et al., 2013*), thus establishing vertex-to-vertex correspondences across individuals and ages. All results were visually inspected to ensure sufficient quality for subsequent analysis. The inner and outer cortical surfaces were used as a constraint to resample the rs-fMRI time courses onto the middle cortical surface with 32,492 vertices using the HCP workbench (*Glasser et al., 2013*), and the time courses were further spatially smoothed on the middle cortical surface with a small Gaussian kernel ($\sigma = 2.55$ mm).

## Generation of fine-grained cortical functional parcellation maps

In this section, we describe detailed steps for generating fine-scaled infant cortical functional parcellation maps (see *Figure 8*). Specifically, we first describe the computation of the local gradient map of FC for each scan, followed by a function-based registration step based on local gradient maps. Then, we detail the computation of both 'age-related' and 'age-independent' parcellation maps based on the functional registration results and our evaluation scheme. Finally, we introduce how we use the parcellation maps to discover the functional network organization development, as well as parcel homogeneity and local efficiency during infancy.

### Computation of individual local gradient map

The local gradient of FC (*Gordon et al., 2016*) identifies sharp changes of RSFC, thus intrinsically representing the transition from one functional parcel to another, and is widely used in generating meaningful fMRI-based cortical parcellations in adult studies (*Gordon et al., 2016*; *Han et al., 2018*; *Schaefer et al., 2018*). For each infant subject, the AP and PA scans are computed separately and only combined during step 8. For each fMRI scan, the computation of the local gradient of FC on the cortical surface is summarized in the following steps. (1) For each fMRI scan, the FC matrix is built by pair-wise correlating the BOLD signal of each vertex with all other cortical vertices in the CIFTI file to create a 32k × 64k RSFC matrix for each hemisphere, where each row represents the RSFC map of a vertex. (2) Each RSFC matrix is transformed to $z$ scores using Fisher's $r$-to-$z$ transformation, which normalizes features of different vertices to a comparable scale. (3) For each fMRI scan, the $z$-transformed RSFC map of each vertex is correlated with all cortical vertices within the same hemisphere, creating a second-order correlation matrix (RSFC-2nd) sized 32k × 32k for each hemisphere, where

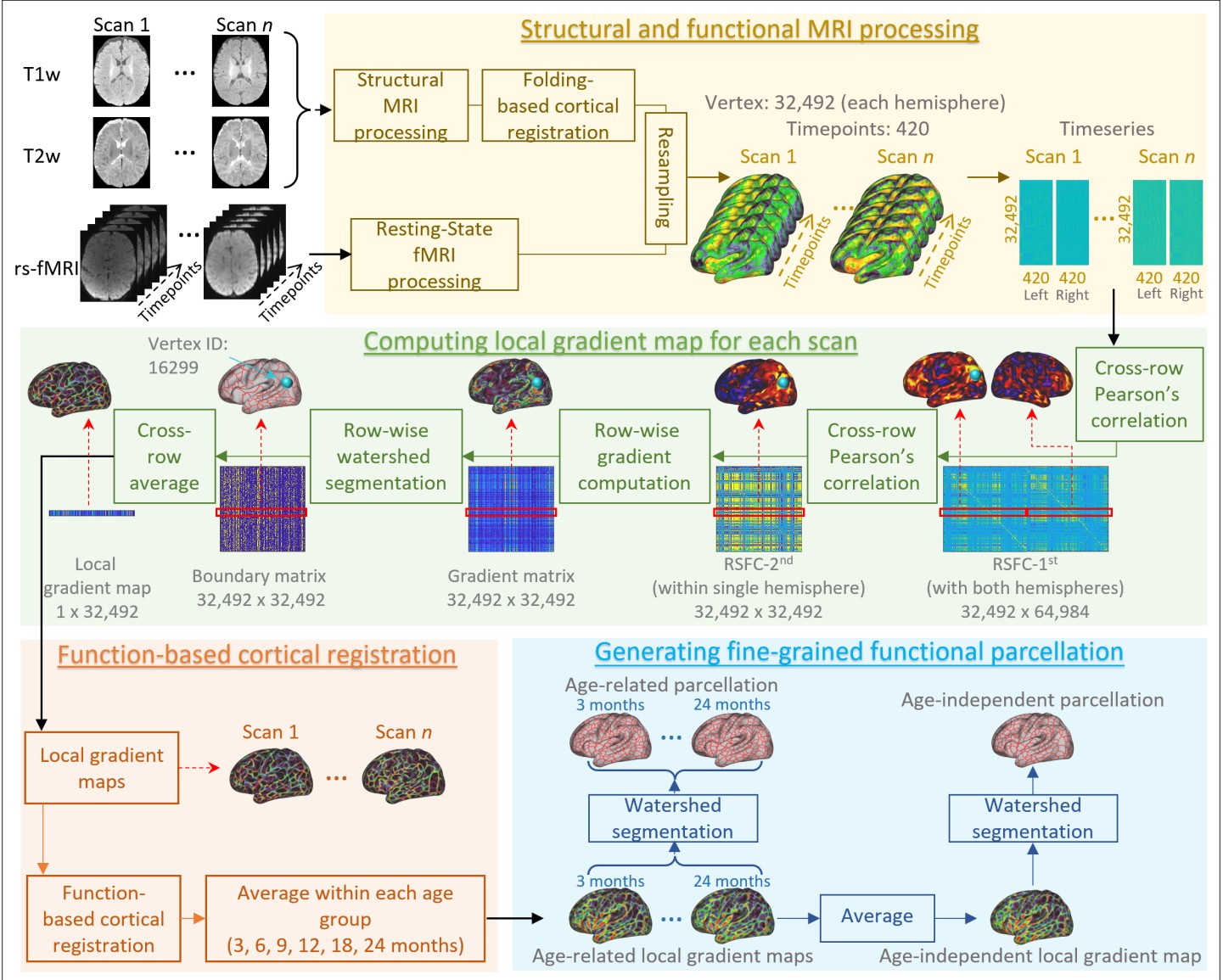

**Figure 8.** An illustration of the procedure of infant cortical parcellation using local gradient map of functional connectivity. Detailed steps are described in the following corresponding parts as: structural and functional MRI processing (sections 'Structural MRI processing', 'Resting-state fMRI processing', and 'Cortical surface reconstruction and mapping'), computing local gradient map of functional connectivity for each scan (section 'Computation of individual local gradient map'), function-based cortical registration (section 'Cortical surface registration based on local gradient map'), and generating fine-grained functional parcellation (section 'Generation of parcellation maps based on local gradient map').

each row represents an RSFC correlation map, based on which the functional transition boundaries can be detected. Note that the RSFC maps (first-order correlation) change smoothly between areas, while the RSFC correlation maps (second-order correlation) exhibit more rapid and abrupt changes, suggesting the clearer presence of a functional boundary between two adjacent areas (*Wig et al., 2014*). (4) For some scan visits consisting of both AP and PA scans, all RSFC-2nd matrices of the same visit from the same subject are averaged so that all subjects contribute equally, even though they may have different numbers of scans in one visit. (5) The gradient of FC is computed on the RSFC-2nd as in *Glasser et al., 2016*, resulting in a 32k × 32k gradient matrix per hemisphere. (6) By performing the watershed-based boundary detection (*Gordon et al., 2016*) on the gradient matrix, we obtain 32k binary boundary maps per hemisphere. (7) The local gradient map is defined as the average of 32k binary boundary maps. (8) The local gradient maps of AP and PA scans are averaged for each session.

When multiple sessions are collected for one visit of a subject, the local gradient maps of different session are also averaged.

## Cortical surface registration based on local gradient map

Previous studies mostly computed population-based local gradient maps, where cortical surfaces were usually co-registered to a common space using only cortical folding-based features. However, due to the highly variable relationship between cortical folds and function, especially in higher-order association regions, researchers have recently become more aware of the necessity of functional feature-based registration (*Coalson et al., 2018*; *Tong et al., 2017*; *Robinson et al., 2018*; *Robinson et al., 2014*; *Nenning et al., 2017*). To this end, in addition to cortical folding-based co-registration, we further use the local gradient of FC as a meaningful functional feature to perform a second-round of co-registration of cortical surfaces for more accurate functional alignment.

Specifically, based on cortical folding-based surface co-registration, (1) the local gradient maps of all scans are averaged to generate the population-mean local gradient map. (2) To improve inter-individual cortical functional correspondences, the local gradient map of each scan is then aligned onto the current population-mean local gradient map using Spherical Demons (*Yeo et al., 2010*) by incorporating local gradient as a feature. (3) All warped functional gradient maps are then resampled and averaged to obtain the newly improved population-mean local gradient map with sharper and more detailed functional architecture. (4) Steps 2 and 3 are repeated iteratively until no visually observed changes in the population-mean local gradient map (four iterations in our experiment). After this procedure, all individual local gradient maps are co-registered, thus establishing functionally relevant cortical correspondences across individuals.

## Generation of parcellation maps based on local gradient map
### Age-related parcellation maps

To capture the spatiotemporal changes of fine-grained cortical functional maps during infancy, we group all scans into six representative age groups, that is, 3, 6, 9, 12, 18, and 24 months of age based on the distribution of scan ages. For each age group, we compute the age-related mean local gradient maps by averaging the local gradient maps of all scans within the group, without any smoothing. Detailed information on each age group is reported in *Table 1*. A watershed method is then applied to each age-related mean local gradient map to generate the corresponding functional parcellation maps (*Gordon et al., 2016*). This watershed segmentation algorithm starts by detecting local minima in 3-ring neighborhoods, and iteratively grows the region until reaching ambiguous locations, where vertices can be assigned to multiple regions. These locations appear to be borders that separate parcels and reflect putative boundaries of FC according to the local gradient maps.

### Age-independent parcellation maps

Ideally, the age-related parcellation maps are the more appropriate representation of the cortical functional architecture at each specific age. However, many neuroimaging studies involve infants of multiple ages, thus the age-related parcellation maps may not be the proper choice due to different parcel numbers and variations in parcel boundaries across ages, thus inducing difficulties in across-age comparisons. To facilitate infant studies involving multiple age groups, we also compute an age-independent local gradient map, which is the average of all six age-related local gradient maps without any smoothing, so that each age group contributes equally to the age-independent map. According to the age-independent local gradient map, we generate the age-independent functional parcellation map using the watershed segmentation method as well. The subsequent parcellation evaluation, functional network architecture, and longitudinal development analyses are performed using the age-independent parcellation maps.

## Evaluation of parcellation maps
### Reproducibility

Ideally, a local gradient-based parcellation map should extract robust common gradient information that shows the transition between parcels. We thus test if the local gradient map is reproducible on different subjects. Therefore, randomly divided 'generating' and 'repeating' groups (*Gordon et al.,*

*2016*; *Schaefer et al., 2018*) are used to calculate the mean local gradient map, separately. These two maps are then binarized by keeping only 25% highest local gradient as in *Gordon et al., 2016*; *Han et al., 2018*, and the dice ratio overlapping index between the two binarized maps is calculated to evaluate the reproducibility of the functional gradient map. This process is repeated multiple times (1000 times in this study) to get a reliable estimation.

## Homogeneity

The local gradient-based parcellation identifies large gradients, representing sharp transitions in FC patterns and avoiding large gradients inside parcels as much as possible. Meanwhile, a parcel that accurately represents a cortical area should not only be distinct from its neighbors in FC pattern, but also has a homogenous FC pattern across all vertices within the parcel. Therefore, we estimate the homogeneity of each parcel as in *Gordon et al., 2016*; *Arslan et al., 2018*. Specifically, we first compute the mean correlation profile of each vertex across all subjects. Next, the correlation patterns of all vertices within one parcel are entered into a principal component analysis; the percentage of the variance that is explained by the largest principal component is used to represent the homogeneity of this parcel.

## Variance

As the FC pattern within a parcel should be relatively uniform, we also measure the variability of the connectivity pattern within each parcel, with smaller variability indicating greater uniformity and hence higher parcellation quality. Specifically, for each parcel, we first obtain a matrix with each column representing the subject-average *z*-score of the FC profile of one vertex in the parcel. Then we compute the sum of the standard deviation of each row to represent the variability of this parcel. The average variability of all parcels is used to represent the variability of the parcellation map.

As parcellation maps usually have different numbers, sizes, and shapes in parcels, to have fair comparisons and be consistent with (*Gordon et al., 2016*; *Han et al., 2018*), we compare our parcellation maps with 'null parcellations'. The null parcellations are generated by rotating by a random amount along the *x*, *y*, and *z* axes on the 32,492 spherical surfaces, which relocate each parcel while keeping the same number and size of parcels. We compare both variability and homogeneity of our parcellation and that of the random rotated null parcellations. Notably, in any random rotation, some parcels will inevitably be rotated into the medial wall, where no functional data exist. The homogeneity/variance of a parcel rotated into the medial wall is not calculated; instead, we assign this parcel the average homogeneity/variance of all random versions of the parcel that were rotated into non-medial-wall cortical regions.

## Variability between *local gradient* maps

A variability map visualizes the variability or dissimilarity between two local gradient maps, and is estimated as follows. For a vertex $v$, $\mathfrak{P}_v$ is defined as a surface patch surrounding $v$ (10-ring neighborhood in this study). For two local gradient maps $g^1$ and $g^2$, two corresponding vectors $p_v^1$ and $p_v^2$ within $\mathfrak{P}_v$ are then extracted. The variability of two local gradient maps at vertex $v$ is computed as $Var\left(g^1, g^2\right) = 0.5 \times \left(1 - corr\left(p_v^1, p_v^2\right)\right)$, where $corr\left(\cdot, \cdot\right)$ stands for Pearson's correlation. As a result, the variability/dissimilarity is within the range of [0, 1], where high values stand for high variability/dissimilarity and vice versa. In this study, we mainly measure the variability between local gradient maps in two aspects: (1) the across-age variability, which computes the variability of local gradient maps between two consecutive age groups to reflect the developmental changes of the local gradient maps; (2) the variability between the age-independent local gradient map and each age-related local gradient map, for quantitatively evaluating whether it is appropriate to use the age-independent parcellation maps for all 6 age groups.

## Functional development analysis

### Functional network detection

To discover the developmental evolution of large-scale cortical functional networks, we employ a network discovery method (*Thomas Yeo et al., 2011*) for each of the six age groups. Specifically, for each subject in each age group, given *n* parcels, we first compute the average time course of each

parcel (excluding the medial wall), and compute the correlation of the average time courses between any two parcels. This results in a $n \times n$ matrix, which is further binarized by setting the top 10% of the correlations to one and the rest to zero to be consistent with *Thomas Yeo et al., 2011*, and because the clustering algorithm appears robust to the choice of threshold. The binarization of connectivity in each individual offers a certain level of normalization so that each subject can contribute the same number of connections, which leads to significantly better clustering results (*Thomas Yeo et al., 2011*). For each age group, all $n \times n$ matrices are averaged across individuals independently. A clustering algorithm (*Lashkari et al., 2010*) is then applied to estimate networks of parcels with similar connectivity profiles.

To determine the optimal cluster number $k$ for each age group, we employ the random split-half test to compute the stability for each $k$, with higher stability corresponding to more meaningful clustering results. Specifically, for each age group, we randomly split all subjects into two folds and run the clustering algorithm separately to obtain two independent clustering results $c_1$ and $c_2$, and the similarity between $c_1$ and $c_2$ is evaluated using the Amari-type distance (*Wu et al., 2016*). This experiment is repeated 200 times for each age group, and the resulting similarities are averaged to represent the stability for $k$. During this process, the range of $k$ is set to [2, 30] according to the existing literature of functional network discovery (*Thomas Yeo et al., 2011*; *Power et al., 2011*).

### Parcel-wise development

We computed the homogeneity and local efficiency of each parcel in the age-independent parcellation to characterize infantile parcel-wise developmental patterns regarding functional homogeneity and functional segregation, respectively. The homogeneity is computed as described in Section "Evaluation of parcellation maps - Homogeneity" for each subject, where higher parcel homogeneity indicates more unified connectivity patterns within the parcel. The local efficiency here represents the node local efficiency (*Shi et al., 2018*; *Eickhoff et al., 2018*) computed using the Gretna Toolkit (*Peng et al., 2023*), that is, 'Gretna_node_local_efficency.m'. It measures the average efficiency of information transfer within local subgraphs and is defined as the inverse of the shortest average path length of all neighbors of a given node among themselves. Formally, it is calculated as $E_{local} = \frac{1}{N_{G_i}(N_{G_i}-1)} \sum_{j,k \in G_i} \frac{1}{L_{j,k}}$, where $N_{G_i}$ represents the number of nodes in the subgraph $G_i$ of node $i$, and $L_{j,k}$ is the average distance (number of steps) between nodes $i$ and $j$ in the subgraph. In the computation, each node represents a parcel, and the subgraph for each parcel is defined as all nodes connected to this parcel with connectivity strength beyond a given threshold. Herein, multiple thresholds are used, keeping 50–5% connections with 1% as a step, and the area under the curve is calculated to represent the local efficiency to avoid the influence of connectivity densities. The local efficiency corresponds to the mean information transfer efficiency between a particular parcel and all its connected nodes, which is proportional to the clustering coefficient. Parcels with higher local efficiency can more effectively share information with their connected parcels, and thus help build effective segregated networks. To show a figure that maps homogeneity and local efficiency onto the cortical surface, and meanwhile shows their temporal development patterns at different ages, we use the sliding window technique to compute homogeneity and local efficiency in each age window by averaging all scans within the same age window. The windows are centered at each month, with a window width of 90 days (±45 days) at 2 months of age, increasing 4 days in width for each following month and reaching 182 days (±91 days) at 2 years of age.

## Acknowledgements

This work was supported in part by NIH grants (MH104324, MH116225, MH117943, MH123202, and MH127544). This work also utilizes approaches developed by an NIH grant (1U01MH110274) and the efforts of the UNC/UMN Baby Connectome Project Consortium.

## Additional information

### Funding

| Funder | Grant reference number | Author |
|---|---|---|
| National Institute of Mental Health | MH116225 | Gang Li |
| National Institute of Mental Health | 1U01MH110274 | Weili Lin |
| National Institute of Mental Health | MH117943 | Li Wang<br>Gang Li |
| National Institute of Mental Health | MH123202 | Gang Li |
| National Institute of Mental Health | MH127544 | Gang Li |

The funders had no role in study design, data collection, and interpretation, or the decision to submit the work for publication.

### Author contributions

Fan Wang, Conceptualization, Software, Formal analysis, Validation, Investigation, Visualization, Methodology, Writing – original draft, Writing – review and editing; Han Zhang, Conceptualization, Data curation, Writing – review and editing; Zhengwang Wu, Data curation, Software, Formal analysis, Writing – review and editing; Dan Hu, Visualization, Writing – review and editing; Zhen Zhou, Data curation, Writing – review and editing; Jessica B Girault, Writing – review and editing; Li Wang, Resources, Data curation, Funding acquisition, Project administration, Writing – review and editing; Weili Lin, Funding acquisition, Project administration, Writing – review and editing; Gang Li, Conceptualization, Resources, Data curation, Formal analysis, Supervision, Funding acquisition, Validation, Investigation, Methodology, Writing – original draft, Project administration, Writing – review and editing

### Author ORCIDs

Fan Wang ![ORCID] https://orcid.org/0000-0002-9955-020X
Gang Li ![ORCID] http://orcid.org/0000-0001-9585-1382

### Decision letter and Author response

Decision letter https://doi.org/10.7554/eLife.75401.sa1
Author response https://doi.org/10.7554/eLife.75401.sa2

## Additional files

### Supplementary files
• Transparent reporting form

### Data availability

All MRI data analyzed during this study are from the UNC/UMN Baby Connectome Project (https://nda.nih.gov/edit_collection.html?id=2848), which is publicly available in NIMH Data Archive, and resulting parcellation maps are released to the public with access through NITRC (https://www.nitrc.org/projects/infantsurfatlas/).

The following previously published dataset was used:

| Author(s) | Year | Dataset title | Dataset URL | Database and Identifier |
|---|---|---|---|---|
| Elison J, Weili L | 2018 | UNC/UMN Baby Connectome Project | https://nda.nih.gov/edit_collection.html?id=2848 | NIMH Data Archive, 2848 |

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
