## [Editor Report]

The study presents a useful fine-grained functional parcellation map of the infant cerebral cortex. The data and methodology presented are solid. It will facilitate future research in neuroanatomy and neurodevelopment disorders.

---

## [Decision Letter]

**Decision letter after peer review:**

Thank you for submitting your article "Generation of Infant-dedicated Fine-grained Functional Parcellation Maps of Cerebral Cortex" for consideration by *eLife*. Your article has been reviewed by 2 peer reviewers, and the evaluation has been overseen by a Reviewing Editor and Tamar Makin as the Senior Editor. The following individuals involved in review of your submission have agreed to reveal their identity: Emma C. Robinson, PhD (Reviewer #1); Kilian M. Pohl (Reviewer #2).

Essential revisions:

1. Include quantitative comparisons to the HCP parcellation.

2. More data on the robustness of network clustering (e.g. binarization/averaging step and random repeat/generation step).

3. Improve discussion (e.g. accessibility to a broader audience, number of parcels, the impact of gender) and references.

4. Improve clarity and readability (methods and Figures).

*Reviewer #1 (Recommendations for the authors):*

As stated this is a very good paper. I think if the comparisons to the HCP and the evaluation of the robustness of the network clustering could be improved it would make a very strong *eLife* paper.

The biggest area for improvement is the clarity of the methods. For example, for aligning the HCP 32k_LR space via fsaverage space, which specific part of reference 53 are they referring to? Assumedly, the authors didn't use landmark registration; therefore, do they mean that they use part of the HCP pipeline to map from fsaverage to FS_LR? If so this doesn't use (53).

For the computation of the individual functional gradient density map more details need to be provided for the motivation behind using the 2nd order correlation matrix and then subsequently correlating to that. Even if it is discussed in previous adult papers this manuscript should stand alone. Some more figures of what these stages look on a cortical surface would help. Unfortunately, I can't clearly follow the process through Figure 1. The resolution of this figure could also be improved.

I also don't fully understand the methods section on 'variability between functional gradient density maps.' What are p1 and p2 here? Are they vectors generated from an image patch surrounding some location?

For section 5.7 'parcel-wise development' I don't follow the description of the efficiency calculation. What is meant be 'the AUC is calculated to represent the local efficiency to avoid the influence of local densities.' What is efficiency here? Is there a calculation for it? I don't have a clear intuition for what it means. What do you mean by 'to have an intuitive and spatiotemporally detailed view of their development'

The method for clustering should be explained. And as stated in the public response the stability of the clustering should be tested and the comparison with the HCP should be quantitatively evaluated.

Some additional points on the results: The analysis of the clustering is quite difficult to follow. Firstly, the colours of the central and peripheral visual clusters are too close. Secondly, I don't understand what is meant by 'the auditory network is distinguished at 3 months and merges into the hand sensorimotor at 6 months' perhaps some arrows would help. I'm also not clear why we have labels for 'hand' sensorimotor' and 'mouth' sensorimotor – is there a reference for this?

Figure resolution needs fixing throughout and Figure 7 needs absolute scale.

References – the paper would benefit from a more thorough citing of the international literature. Pg 19 references only a few papers that estimate resting state networks for babies and ignores:

Doria, Valentina, et al. "Emergence of resting state networks in the preterm human brain." Proceedings of the National Academy of Sciences 107.46 (2010): 20015-20020.

Fitzgibbon, Sean P., et al. “The developing Human Connectome Project (dHCP) automated resting-state functional processing framework for newborn infants.” NeuroImage 223 (2020): 117303.

The references for FLIRT are:

M. Jenkinson and S.M. Smith. A global optimisation method for robust affine registration of brain images. Medical Image Analysis, 5(2):143-156, 2001.

M. Jenkinson, P.R. Bannister, J.M. Brady, and S.M. Smith. Improved optimisation for the robust and accurate linear registration and motion correction of brain images. NeuroImage, 17(2):825-841, 2002.

Reference 48 is clearly heavily inspired by FSL's FIX and therefore FIX should be referenced.

G. Salimi-Khorshidi, G. Douaud, C.F. Beckmann, M.F. Glasser, L. Griffanti S.M. Smith.

Automatic denoising of functional MRI data: Combining independent component analysis and hierarchical fusion of classifiers. NeuroImage, 90:449-68, 2014

L. Griffanti, G. Salimi-Khorshidi, C.F. Beckmann, E.J. Auerbach, G. Douaud, C.E. Sexton, E. Zsoldos, K. Ebmeier, N. Filippini, C.E. Mackay, S. Moeller, J.G. Xu, E. Yacoub, G. Baselli, K. Ugurbil, K.L. Miller, and S.M. Smith. ICA-based artefact removal and accelerated fMRI acquisition for improved resting state network imaging. NeuroImage, 95:232-47, 2014

The citation for Melodic is also missing.

Beckmann, Christian F., and Stephen M. Smith. "Probabilistic independent component analysis for functional magnetic resonance imaging." IEEE transactions on medical imaging 23.2 (2004): 137-152.

For functionally driven cortical surface registration (pg 26) the authors should also reference MSM and other papers that have previously used spherical demons to drive functional alignment.

Robinson, E. C., K. Garcia, M. F. Glasser, Z. Chen, T. S. Coalson, A. Makropoulos, J. Bozek, R. Wright, A. Schuh, M. Webster, J. Hutter, A. Price, L. Cordero Grande, E. Hughes, N. Tusor, P. V. Bayly, D. C. Van Essen, S. M. Smith, A. D. Edwards, J. Hajnal, M. Jenkinson, B. Glocker, and D. Rueckert. 2018. "Multimodal Surface Matching with Higher-Order Smoothness Constraints." NeuroImage 167. doi:10.1016/j.neuroimage.2017.10.037.

Robinson, Emma C., Saad Jbabdi, Matthew F. Glasser, Jesper Andersson, Gregory C. Burgess, Michael P. Harms, Stephen M. Smith, David C. Van Essen, and Mark Jenkinson. 2014. "MSM: A New Flexible Framework for Multimodal Surface Matching." NeuroImage 100: 414-26. doi:10.1016/j.neuroimage.2014.05.069.

Nenning, Karl-Heinz, Hesheng Liu, Satrajit S. Ghosh, Mert R. Sabuncu, Ernst Schwartz, and Georg Langs. 2017. "Diffeomorphic Functional Brain Surface Alignment: Functional Demons." NeuroImage. Elsevier.

When discussing parcellation validation and homogeneity Arslan et al. 2018 should be cited.

Arslan, Salim, et al. "Human brain mapping: A systematic comparison of parcellation methods for the human cerebral cortex." NeuroImage 170 (2018): 5-30.

*Reviewer #2 (Recommendations for the authors):*

Figures: overall the figures are difficult to interpret and the caption fail to explain nomenclature and provide a take home message for the corresponding figure, i.e., figures cannot be understood without carefully reading the text. In the detail comments below – comments for one figure also apply to proceeding ones.

– Figure 1: Given the amount of detail provided, it should be moved to the supplement or method section where the individual components are then described.

– Figure 2: Instead of a)-d) it would be fantastic if one provides short heading (such as Proposed Atlas). Also, at a first glance they do not seem to have anything to do with each other (other than c and d – but then it is not clear why there are three images in d). d) might be better for the supplement as the message of Figure 2 is unclear.

– Figure 3: what is L and R, why are both hemispheres shown even though the result section does not discuss differences in hemispheres? Why are roughly 450 parcellations chosen for each hemisphere – why is that better than e.g. 600 or 300 parcellations? What is 3M … 24M referring to and which brains belong to which group? I know it is somewhere described in the text but each figure should be self-explanatory.

Figure 4 - b) is never explained in the text.

Figure 5: why are hemisphere not oriented as in other pictures?

Figure 6: chosen stability criteria seems arbitrary – why not chose max?

Point of Figure 8 is entirely unclear.

The result section lacks focus. For example, it jumps back and forth between "common" and age-related atlas. The reason for the common atlas is not made clear until the method section and Section 2.3 seems to indicate that the age-related atlas are not needed at all.

It would be helpful if the regions and networks mentioned in the text would also be pointed out in (one) figure so that the manuscript can actually be used as a self-contained reference.

Frequent use of jargon/uncommon terms: group-average functional gradient, functional gradient density maps, Temporal variabilities, the functional architecture development, "Keeping this spatial distribution".

Certain statements are not supported:

– "dice ratio reaches 0.9295 {plus minus} 0.0021 indicating the high reproducibility" – what is high and low scores for this application?

– "demonstrates that our method can well capture these important and detailed functional gradient patterns, which are usually missed by the compared methods" Output of compared method is not shown.

– how are visual areas V1, MT, MST, sensorimotor areas 2, 3, 4, auditory areas A1, LBelt, and language areas 44, 45 are defined (in infants)?

Regarding method section:

– alignment to HCP template (generated from adults) needs to be justified – is that a potential source of error/how do they relate to infant brains.

– Table 1 should be presented at beginning of the method section – can you provide % of sex in each age bracket. Also the % seems to be significantly different between age groups – how does that impact the atlas?

[Editors’ note: further revisions were suggested prior to acceptance, as described below.]

Thank you for resubmitting your work entitled "Infant-dedicated Fine-grained Functional Parcellation Maps of Cerebral Cortex" for further consideration by *eLife*. Your revised article has been evaluated by Tamar Makin (Senior Editor) and a Reviewing Editor.

The manuscript has been improved but there are some remaining issues that need to be addressed, as outlined below:

*Reviewer #3 (Recommendations for the authors):*

I still believe that the proposed atlas could be of great importance to the field. However, the rather sloppy responses to my concerns lowers my overall enthusiasm (many issues were not responded to or they were responded to but the manuscript was not altered).

I originally stated that "Diminishing enthusiasm is the lack of focus in the result section, the frequent use of jargon, and figures that are often difficult to interpret." which still holds true.

For example, the abstract and intro was hardly edited (only expanded – see blue highlights) so that these sections (like the rest of the manuscript) are difficult to read. For example, the term "functional gradient density" is not a common term in the community (and was not even addressed in the response) and the meaning behind "age-common fine-grained parcellation maps" or "infant brain functional developmental maps" is difficult to grasp. While many of the figures improved, the take home message of Figure 2 -6 is unclear as is the reason for showing Figure 7 at all.

Parts added to the article bring up additional concerns. Table 1 clearly shows that the greater the imbalance between sexes, the greater the impact on reproducibility (up to 8% changes). The criteria for reproducibility (based on thresholding) does not seem to relate to how the functional parcellations maps are generated.

The added sentence:

"the across-age variability shows a multi-peak fluctuation, … again for 18-24 months."

might again be related to sex differences as it seems to correlate with those and are biologically not well supported.

The intuition behind null parcellations (which are not explained until much later) is missing as I assume that the measured deviations partly depend on how much the vertices are perturb (what does the x-axis in the plot of Figure 4 refer to).

The selected number of networks based on the plots in Figure 5a still seems arbitrary.

Something I missed in the initial review (I apologize) is the use of PA and AP scans. Why are they not combined before performing functional analysis to correct for spatial distortion? Why include scan sessions that do not contain both scans? How are differences across acquisition sites modeled?

Given that this is an atlas hopefully used by many other studies, a more detailed description of the demographics of each age group would be helpful. While mentioned several times, what is the web address of the atlas and are there any restrictions for downloading it?

---

## [Author Response]

Essential revisions:1. Include quantitative comparisons to the HCP parcellation.

Great suggestion! We added quantitative comparisons using the Hausdorff distance, and the results are listed in Figure 4 in the revised manuscript. Similar to the comparison in parcel variance and homogeneity, the 1,000 null parcellations were created by randomly rotating our parcellation with small angles on the spherical surface 1,000 times. We compared our parcellation and the null parcellations by accordingly evaluating their Hausdorff distances to some specific areas of the HCP parcellation on the spherical space, including Brodmann's area 2, 3b, 4+3a, 44+45, V1, and MT+MST.

From the results, we can observe that our parcellation generally shows statistically much lower Hausdorff distances to the HCP parcellation, suggesting that our parcellation generates parcel borders that are closer to the HCP parcellation boundaries compared to the null parcellations. However, we noticed very few null parcellations that show smaller Hausdorff distances compared to our parcellation. A possible reason comes from our surface registration process with the HCP template purely based on cortical folding, without using functional gradient density maps, which are not available in the HCP template. As a result, this does not ensure high-quality functional alignment between our infant data and the HCP space, thus inevitably increasing the Hausdorff distance between our parcellation and the HCP parcellation.

2. More data on the robustness of network clustering (e.g. binarization/averaging step and random repeat/generation step).

Excellent point! In the revision, we added two experiments to test the robustness of network clustering, e.g., compared stability of ‘binarizing first’ and ‘averaging first’, and used bootstrapping and voting for clustering, which are detailed as follows:

– For the step of “binarizing before averaging”, we followed the method proposed by Yeo et al. (5). In this method, all correlation matrices are binarized according to the individual-specific thresholds. Specifically, each individual-specific threshold is determined according to the percentile, and only 10% of connections are kept and set to 1, while all other connections are set to 0. Yeo et al. (5) explained their motivation for doing so as “the binarization of the correlation matrix leads to significantly better clustering results, although the algorithm appears robust to the particular choice of the threshold”. We consider that the possible reason is that the binarization of connectivity in each individual offers a certain level of normalization so that each subject can contribute the same number of connections. If averaging occurs before binarizing, the actual connectivity contributed by different subjects could be very different, which leads to bias. Meanwhile, we tested the stability of ‘binarizing first’ and ‘averaging first’, and the result is shown in Author response image 1. This figure suggests a similar conclusion as (5), where binarizing first before averaging leads to better clustering stability.

**Author response image 1. sa2fig1:** The comparison of clustering stability of different methods. The red line refers to the clustering stability when binarizing the correlation matrices first and then averaging the matrices across individuals, while the blue line refers to the clustering stability when averaging the correlation matrices across individuals first and then binarizing the average matrix.

– For the final clustering results, we performed our clustering method using bootstrapping 100 times, and the final result is a majority voting of each parcel. The comparison of these two results is shown in Author response image 2. Overall, we do observe good repeatability between these two results. However, we also observed that few parcels show different patterns between the two results, which are mostly spatially located around the boundaries of networks or the medial wall. The pattern of the observation that “the posterior frontoparietal expands to include the parahippocampal gyrus from 3-6 months and then disappears at 9 months – remains” was not repeated in the bootstrapped results. These results might suggest that the clustering method is quite robust, the discovered patterns are relatively stable, and the differences between our original results and bootstrapping results might be caused by noises or inter-subject variabilities.

**Author response image 2. sa2fig2:** Top panel: the network clustering results using all data in the original manuscript. Bottom panel: the network clustering results using majority voting through 100 times of bootstrapping. Black circles and red arrows point to the parahippocampal gyrus, which was included in the posterior frontoparietal network, and is not well repeated in the bootstrapped results. (M: months).

3. Improve discussion (e.g. accessibility to a broader audience, number of parcels, the impact of gender) and references.

– For the number of parcels, even without performing precise cortical surface registration based on fine-grained functional features, recent adult fMRI-based parcellations greatly increased parcel numbers, such as up to 1,000 parcels in Schaefer et al. (9), 518 parcels in Peng et al. (20), and 1,600 parcels in Zhao et al. (21). Their parcel numbers and accuracy could be further improved when adopting our method for precise cortical surface registration using fine-grained functional features. For infants, we do agree that the infant functional connectivity might not be as strong as in adults. However, there are opinions (22-24) that the basic units of functional organization are likely to present in infant brains, and brain functional development gradually shapes the brain networks. Therefore, the functional parcel units in infants could be possibly on a comparable scale to adults. We added this in the Discussion section.

– For references, we added 20 references in this revision. Their IDs in the revised manuscript are: 16, 17, 20, 21, 25, 26, 27, 28, 29, 50, 51, 52, 53, 54, 61, 62, 63, 73, 74, and 75.

– To improve the accessibility to a broader audience, we reformulated some uncommon terms. For example, we changed “Functional-architecture development” to “functional network development”; replaced “Temporal variabilities” to “Across-age variabilities”; replaced “Group-average functional gradient density” to “mean functional gradient density”, or “3-month mean functional gradient density”, or “age-specific mean functional gradient density”; and replaced “Keeping this spatial distribution” to “While exhibiting this overall spatial distribution, the across-age variability shows a multi-peak fluctuation …”.

– For the impact of gender, we added an experiment by rebalancing the sex for each group and testing the consistency between our results and the rebalanced results. Experimental results show that more than 91% of strong gradient densities are repeated with the threshold of top 25%, and more than 95% of gradients are repeated with the threshold of 50%. The results are detailed in Author response table 1.

**Author response table 1. sa2table1:** Reproducibility of gradient density maps by rebalancing the sex in each age group (M: months).

Threshold	3 M	6 M	9 M	12 M	18 M	24 M
Top 25%	95.25%	97.00%	96.73%	91.95%	94.96%	95.15%
Top 50%	96.70%	97.80%	97.37%	95.30%	97.32%	97.33%

4. Improve clarity and readability (methods and Figures).

– For methods, we added more details, including the procedure of cortical surface registration, especially for mapping cortical surfaces from individual spaces to the HCP ‘fs_LR’ space. Meanwhile, we improved the method flowchart by integrating more small figures on the cortical surface and updated figure details. We also added more details in the calculation of variability between functional gradient density maps, and the calculation of parcel local efficiency.

– For figures, we thoroughly improved figure quality by replacing them with high-quality figures, and we also added descriptive short head in Figure 1, explanatory captions for Figure 4 and Figure 8, added arrows pointing to important network changes in Figure 5, and explained abbreviations in the captions of all figures and tables (such as M: month, L: left hemisphere, R: right hemisphere, etc.).

[Editors’ note: what follows is the authors’ response to the second round of review.]Reviewer #3 (Recommendations for the authors):I still believe that the proposed atlas could be of great importance to the field. However, the rather sloppy responses to my concerns lowers my overall enthusiasm (many issues were not responded to or they were responded to but the manuscript was not altered).

We thank the reviewer for the careful and insightful review of our manuscript and parcellations. We do recognize that there are some points that could be strengthened. Accordingly, we mainly made the following corrections in this revision:

1. Regenerating the parcellation maps after balancing gender, and AP/PA within scan sessions and updated all related results.

2. Reformulating some expressions for easy understanding.

3. Rewriting the Abstract and Introduction, and thoroughly edited the rest of the paper with a neuroscientist.

I originally stated that "Diminishing enthusiasm is the lack of focus in the result section, the frequent use of jargon, and figures that are often difficult to interpret." which still holds true.For example, the abstract and intro was hardly edited (only expanded – see blue highlights) so that these sections (like the rest of the manuscript) are difficult to read. For example, the term "functional gradient density" is not a common term in the community (and was not even addressed in the response) and the meaning behind "age-common fine-grained parcellation maps" or "infant brain functional developmental maps" is difficult to grasp.

We thank the reviewer for this comment. We rewrote our Abstract and Introduction with the help of a neuroscientist and changed some uncommon expression. We changed ‘functional gradient density’ to ‘local gradient of functional connectivity’ or ‘local gradient map’ throughout this revision, changed ‘age-common fine-grained parcellation maps’ to ‘age-independent parcellation maps’, changed ‘age-specific’ to ‘age-related’, and changed ‘infant brain functional developmental maps’ to ‘developmental maps of infant brain functional connectivity’. We hope these changes will help better understand our paper.

While many of the figures improved, the take home message of Figure 2 -6 is unclear as is the reason for showing Figure 7 at all.

For Figure 7 (now Figure 6), other than a simple trajectory, we are showing developmental maps, where we not only observe the development trend (e.g., monotonic decrease or multi-peak fluctuation), but also intuitively see their detailed spatial distributions (e.g., whether the unimodal cortex is higher or not). This is quite new compared to previous studies, which typically display a chart including several lines. We added take-home messages in the figure captions for Figures 1, 3, and 6.

Parts added to the article bring up additional concerns. Table 1 clearly shows that the greater the imbalance between sexes, the greater the impact on reproducibility (up to 8% changes). The criteria for reproducibility (based on thresholding) does not seem to relate to how the functional parcellations maps are generated.

We do agree with the reviewer that imbalanced gender probably introduces additional bias to functional parcellation maps. In this revision, we revisited our subjects, removed subjects or scan sessions that only have AP or PA scans, and balanced the gender. Our newly generated parcellation maps are now based on subjects with balanced gender and AP/PA scans. Details about our subjects in each age group are shown in

Table 1. We accordingly updated all related results in the revision.

The added sentence:"the across-age variability shows a multi-peak fluctuation, … again for 18-24 months."might again be related to sex differences as it seems to correlate with those and are biologically not well supported.

We do agree with the reviewer that the multi-peak fluctuation shown in brain functional development is not well supported in the literature. After rebalancing subjects in terms of sex and AP/PA scans, the multi-peak fluctuation is still shown. However, the underlying mechanisms of such developmental patterns remain to be further investigated, and further studies and data are necessary to reproduce these patterns. To explain the current results, besides brain development, this might be related to inter-subject variabilities, as the BCP is designed as an accelerated longitudinal study, where subjects at different timepoints typically are not the same, which might induce inter-subject variabilities and affect the development patterns. We added this between line 349 and line 353 in the revised manuscript.

The intuition behind null parcellations (which are not explained until much later) is missing as I assume that the measured deviations partly depend on how much the vertices are perturb (what does the x-axis in the plot of Figure 4 refer to).

We thank the reviewer for this comment. To evaluate a parcellation's homogeneity, it is essential to compare it with a null model. This comparison not only focuses on how homogeneous the parcels are, but also on whether they are more homogeneous than what would be expected from parcels randomly placed with the same size and shape. Therefore, we assessed the degree to which a parcellation was more homogenous than a null model consisting of many parcellations with randomly placed parcels of the same size, shape, and relative position to each other. We hope this explains why the null parcellation is introduced in the comparison, and we added this explanation to the null parcellation between line 196 to line 201 in the revised manuscript. Moreover, we randomly rotated the parcellation map with a degree between 0 an 0.1*pi. This experiment follows previous functional parcellation studies (1-3) for convenient comparison. In Figure 4, for each circle, the y coordinate represents its corresponding homogeneity, and the x coordinate is set to an integer between 1 and 50 in sequence. The purpose is to evenly distribute all gray circles in an x-range so that most circles can be seen. We added a brief description of the x-axis in the caption of Figure 4.

The selected number of networks based on the plots in Figure 5a still seems arbitrary.

We agree with the reviewer that the selection of networks seems arbitrary, probably due to a lack of explanation. Empirically, it is difficult to determine the best cluster number, especially when looking for meaningful network patterns. Herein, the clustering number (network number) was chosen under several considerations. First, the stability is computed as in *(5, 30)*, which is used as a reference where a higher stability is assumed to represent a better network number. Then, three empirical factors are considered: 1) for most studies, a coarse network clustering falls around 10 networks, 2) the network number is likely to grow with age, and 3) the network patterns are relatively consistent between two consecutive age groups. By considering both the stability and three empirical factors, we reviewed all clustering results between 5 and 15 clusters and obtained the numbers reported in the manuscript. We added this detailed explanation between line 214 and line 222 in the revised manuscript.

Something I missed in the initial review (I apologize) is the use of PA and AP scans. Why are they not combined before performing functional analysis to correct for spatial distortion? Why include scan sessions that do not contain both scans?

We thank the reviewer for this comment. First, for this revision, we removed those scans that only contain AP or PA scans to avoid possible bias. Second, for the spatial distortion, we performed EPI geometric distortion corrections for AP and PA scans using *topup* from FSL (*6*) and verified the distortion correction results by visual inspection. This is explained in more detail between line 451 and 453 in the revised manuscript. To combine information from the AP and PA scans, one usually either directly concatenates the BOLD signals of AP and PA, or averages the resultant metrics maps derived from AP and PA. Considering possible misalignment between AP and PA scans even after the correction, we preferred not to directly concatenate the BOLD signals of corresponding voxels, for that will cause spurious correlation and introduce artifacts (*7*). Instead, in this revision, the AP and PA scans of each session were combined by averaging their local gradient maps. Different scan sessions of the same visit were then averaged if more than one scan was collected from the same subject during one visit. We reformulated the method and related descriptions in the revised version to make the steps clearer, which can be found between lines 498 to 499, and lines 512 to 520 of the revised manuscript.

How are differences across acquisition sites modeled?

We thank the reviewer for mentioning this important point. According to this comment, we separated subjects obtained by different sites, generated parcellation maps separately, and showed the resulting parcellation maps in Author response image 3. We overlay two parcellation maps over each other, and we can observe that they largely resemble each other. The reasons can be twofold. First, for the data acquisition, two scanners from two sites are calibrated by phantoms. Second, the same scanning protocol is used to minimize the inter-site data inhomogeneity. Previous fMRI-based analyses based on the Baby Connectome Project also did not report site differences. However, we do acknowledge that there might exist possible minor biases brought by different sites. For future studies that model the developmental trajectory, we would consider site differences in the statistic model.

**Author response image 3. sa2fig3:** Parcellation boundaries from different sites. The black color indicates the parcel boundaries agreed by both sites, and the red and yellow colors show the boundaries from UNC and UMN, respectively. The parcellation boundaries from the two sites are observed very close to each other, which suggests that acquisition sites do not place strong influences on the parcellation map.

Given that this is an atlas hopefully used by many other studies, a more detailed description of the demographics of each age group would be helpful.

We do agree with the reviewer that detailed demographics would be helpful for future users of the parcellation maps. Now for each age group, we have the subject number, sMRI scan number, fMRI scan number, sex, scan age, and gestational age listed in Table 1. Besides, Figure 7 gives an intuitive demonstration of the scan schedule for different subjects.

While mentioned several times, what is the web address of the atlas and are there any restrictions for downloading it?

We thank the reviewer for this suggestion. We made our parcellations and networks publicly available on both NITRC (https://www.nitrc.org/projects/infantsurfatlas/) and BALSA (https://balsa.wustl.edu/study/88638) with source files of all figures. Our atlases can be downloaded for free without any restrictions. This is now pointed out in the revised manuscript by adding a footnote on page 4.

1. E. M. Gordon, T. O. Laumann, B. Adeyemo, J. F. Huckins, W. M. Kelley, S. E. Petersen, Generation and evaluation of a cortical area parcellation from resting-state correlations. *Cerebral cortex* 26, 288-303 (2014).

2. A. Schaefer, R. Kong, E. M. Gordon, T. O. Laumann, X.-N. Zuo, A. J. Holmes, S. B. Eickhoff, B. T. J. C. C. Yeo, Local-global parcellation of the human cerebral cortex from intrinsic functional connectivity MRI. 28, 3095-3114 (2018).

3. L. Han, N. K. Savalia, M. Y. Chan, P. F. Agres, A. S. Nair, G. S. J. C. C. Wig, Functional Parcellation of the Cerebral Cortex Across the Human Adult Lifespan. 28, 4403-4423 (2018).

4. F. Wang, C. Lian, Z. Wu, H. Zhang, T. Li, Y. Meng, L. Wang, W. Lin, D. Shen, G. Li, Developmental topography of cortical thickness during infancy. *Proceedings of the National Academy of Sciences* 116, 15855-15860 (2019).

5. A. Sotiras, J. B. Toledo, R. E. Gur, R. C. Gur, T. D. Satterthwaite, C. Davatzikos, Patterns of coordinated cortical remodeling during adolescence and their associations with functional specialization and evolutionary expansion. *Proceedings of the National Academy of Sciences* 114, 3527-3532 (2017).

6. S. M. Smith, M. Jenkinson, M. W. Woolrich, C. F. Beckmann, T. E. Behrens, H. Johansen-Berg, P. R. Bannister, M. De Luca, I. Drobnjak, D. E. Flitney, Advances in functional and structural MR image analysis and implementation as FSL. *Neuroimage* 23, S208-S219 (2004).

7. W. Jiang, H. Zhang, Y. Wu, L. Hsu, D. Hu, D. Shen, in *MICCAI 2019*. (Springer, Shenzhen, China, 2019).